# A Personalized Conversational Benchmark: Towards Simulating Personalized Conversations

## Abstract

We present PERSONACONVBENCH, a large-scale benchmark for evaluating personalized reasoning and generation in multi-turn conversations with large language models (LLMs). Unlike existing work that focuses on personalization or conversational structure in isolation, PERSONACONVBENCH tightly integrates both, offering three core tasks: sentence classification, impact regression, and user-centric text generation, covering 10 diverse Reddit-based domains. This design enables systematic analysis of how personalized conversational context can shape LLM outputs in realistic, multi-user conversational scenarios. We systematically benchmark several commercial and open-source LLMs under a unified prompting setup, and observe that incorporating personalized conversational history yields substantial performance boosts—e.g., achieving a 198% relative gain over the best non-conversational baseline in sentiment classification. By releasing PERSONACONVBENCH with comprehensive evaluations and codes, we aim to facilitate research on LLMs that can adapt to individuals' conversational styles, track long-term context, and generate more contextually rich and engaging responses.

**Codes:** `https://anonymous.4open.science/r/PERSONA-BENCH/README.md`

## 1 Introduction

Personalization is crucial for developing language models that can learn and adapt to individual users' unique preferences, communication styles, and interaction histories, ultimately generating more tailored and effective outputs (Zhang et al., 2024; Wu et al., 2024). Recent works, such as LaMP (Salemi et al., 2024), LongLaMP (Kumar et al., 2024), and PGraphRAG (Au et al., 2025), provide benchmarks focused on single-turn or long-form generation tasks using user-specific data, serving as valuable testbeds for modeling user behavior in dynamic contexts. Given the conversational nature of various applications, including social platforms, virtual assistants, customer support, and collaborative tools, multi-turn exchanges with evolving context and intent provide an ideal setting for investigating personalization, as they mirror real-world interactions that require language models to adapt and respond accordingly.

Prior personalization work mostly treats each user utterance as *independent*, *e.g.*, LaMP leverages the set of all reviews written by a user and then uses all these reviews to generate a personalized title for a new review (Salemi et al., 2024).

Conversely, most multi-turn conversation work focuses on modeling interaction structure or dialogue coherence while remaining largely user-agnostic. To date, there is no benchmark that jointly captures the challenges of personalization and multi-turn conversational structure. We defer the detailed discussion of current works to Appendix A.

**Our Proposal**. We address this gap by introducing PERSONACONVBENCH, the *first* personalized conversational benchmark for multi-turn conversations.

PERSONACONVBENCH covers 10 diverse domains and includes 19,215 posts encompassing over 111,239 conversations from 3,878 users, as detailed in Table 1 and Table 2. Within each domain, we instantiate three canonical evaluation paradigm—classification, regression, and conversational text generation-yielding a total of $10 \times 3 = 30$ distinct dataset configurations that jointly capture broad domain and task diversity. Our benchmark provides a basis for both researchers and practitioners to

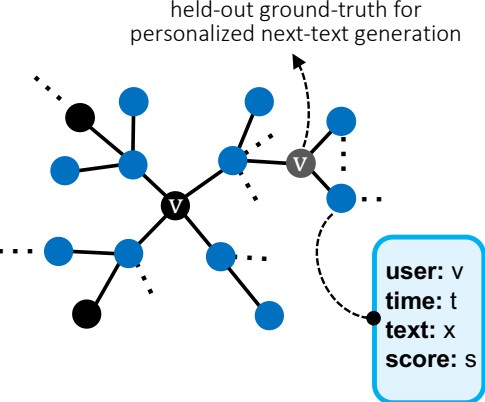

Figure 1: Illustration of the personalized conversational setting in PERSONACONVBENCH. The center black node represents a user $v$ who initiates a post, leading to multiple conversational trajectories as other users respond. Over time, user $v$ replies to some of these users, forming deeper branches in the graph. For each reply made by user $v$, we can construct a prediction task—either classifying the response's sentiment, forecasting its community score, or generating the response text—based on the earlier parts of the same trajectory and additional user-specific history. These tasks rely on realistic multi-user, multi-turn settings with graph-structured conversational data. Each message is annotated with username $v$, timestamp $t$, message content $x$, and feedback score $s$, supporting fine-grained personalization across all task types.

study and develop techniques for a variety of new tasks and settings that our benchmark gives rise to.

Additionally, recent advances in large language models (LLMs) show the importance of evaluating how these powerful generative models incorporate user-specific context in multi-turn dialogue (Yi et al., 2024).

Unlike prior benchmarks restricted to dyadic (user–agent) dialogues, our setup captures complex, multi-user interactions that arise naturally in online forums. Each conversation is represented as a graph of user utterances, as illustrated in Figure 1, where user-specific response trajectories can be extracted for training and evaluation. This design enables rigorous analysis of how personalized context and user history influence language models' outputs under realistic multi-turn conditions.

**Main Findings**. Our experiments demonstrate the effectiveness of our personalized conversational benchmark.

Notably, we find that by carefully leveraging a set of user-specific conversations, we can better personalize the generations and responses of these large models, leading to significant performance gains across all 3 tasks and 10 diverse domains.

In particular, we observe a *198% improvement in classification*, an *11.5% gain in regression*, and a *35.1% gain in conversational text generation* compared to baselines that do not incorporate personalized multi-turn data.

The key contributions of this work are as follows:

- **Problem Formulation:** We formulate the problem of personalized conversation generation, and investigate a few different personalized conversational tasks, including classification, regression, and a more advanced generation task.

- **Benchmark:** We propose a novel benchmark for personalized conversational tasks that include a few different fundamental tasks including classification, regression, and conversational generation, and for each of these personalized conversational tasks, we investigate a highly diverse set of domains, notably, we focus on ten diverse domains for each personalized conversational task (*i.e.*, classification, regression, and generation).

- **Effectiveness:** We conduct extensive experiments on our large-scale personalized conversational benchmark with various LLMs, consisting of 3 tasks (classification, regression, and generation),

| Conversational Domain | Conversation Style | Conversational Engagement | Conversation Purpose | User Interactivity |
|---|---|---|---|---|
| Worldnews | | Debate-Driven | Education | Low |
| Science | FORMAL | Debate-Driven | Education | Medium |
| Politics | | Debate-Driven | Education | Low |
| Technology | | Information Sharing | Education | Medium |
| Gaming | | Community-Based | Entertainment | High |
| Life | CASUAL | Community-Based | Socializing | Medium |
| Movies | | Opinion-Based | Entertainment | Medium |
| Books | | Opinion-Based | Entertainment | High |
| Entrepreneur | MOTIVATIONAL | Supportive | Advice/Support | High |
| Art | CREATIVE | Community-Based | Entertainment | Medium |

Table 1: Ours includes diverse conversational data from a wide range of domains, conversation styles, engagement styles, interaction styles, conversational purpose, and user interactivity levels.

across 10 diverse domains (*e.g.*, different conversational styles, engagement patterns, level of interactivity, etc), and a wide variety of metrics (*e.g.*, for personalized conversational generation, we show ROUGE (Lin, 2004), METEOR (Banerjee & Lavie, 2005), BLEU (Papineni et al., 2002), SBERT similarity (Reimers & Gurevych, 2019)).

## 2 RELATED WORK

### 2.1 PERSONALIZATION IN LLMS

Benchmarks such as LaMP (Salemi et al., 2024) and LongLaMP (Kumar et al., 2024) evaluate personalization in classification and long-form generation tasks, while PersoBench (Afzoon et al., 2024) and Persona-SQ (Lin et al., 2024) extend evaluation to persona consistency and scalable question suggestion. PersonalLLM (Zollo et al., 2025) shifts toward alignment datasets capturing diverse preferences beyond predefined personas. Recent methods adapt LLMs using personalized PEFT (Tan et al., 2024), low-rank preference modeling (Bose et al., 2025), multistage writing frameworks (Li et al., 2023), and conversational health agents (Abbasian et al., 2023). Evaluation efforts include metrics like EGISES (Patel et al., 2024). Together these works highlight growing interest in efficient and generalizable personalization (Chen et al., 2024).

### 2.2 CONVERSATIONS

Multi-turn benchmarks typically assess coherence, commonsense, and multilingual dialogue without personalization (Kwan et al., 2024; Zhao et al., 2023; Mendonça et al., 2024; Zhang et al., 2023; Castillo-Bolado et al., 2024). Recent studies emphasize simulating user-specific conversational trajectories, especially with sparse feedback (Ashok & May, 2024), motivated by the limits of human data and the rise of synthetic dialogue generation (Villalobos et al., 2024; Zhang et al., 2024). While simulation enables scalable alignment, it risks collapse and poor generalization (Shumailov et al., 2024), underscoring the need for personalization-aware conversational benchmarks (Zhang et al., 2025; Zhao et al., 2025).

A detailed discussion is provided in Appendix A.

## 3 PERSONALIZED CONVERSATIONAL BENCHMARK: SETUP AND TASKS

**Problem Formulation.** We introduce a benchmark for *personalized conversational understanding and generation*, where models must produce outputs $y$ that are both contextually relevant and tailored to a specific user's history. Let $\mathcal{U} = \{u_1, \ldots, u_N\}$ denote users, each with a trajectory set $\mathcal{C}_u$. For dataset $\mathcal{T} = \{(f_1, y_1, \mathcal{C}_{u_1}), \ldots, (f_M, y_M, \mathcal{C}_{u_M})\}$, the task is to generate $\bar{y}$ from input $f$ and $\mathcal{C}_u$, requiring reasoning over both immediate context and user-specific intent and style.

**Benchmark Overview.** The benchmark evaluates whether *LLMs* can generate personalized outputs from prompts $x$ and histories $\mathcal{C}_u$. It spans 10 conversational domains and defines three tasks: (i)

| Domain | Posts | User | Conversation | | CtxLen | | Depth | Width | UsrInt |
|---|---|---|---|---|---|---|---|---|---|
| | | | Avg. | Tot. | Avg. | Med. | Avg. | Avg. | Avg. |
| Art | 1 008 | 191 | 5.07 | 5 113 | 210.64 | 109 | 6.15 | 5.87 | 7.67 |
| Books | 824 | 184 | 6.96 | 5 735 | 166.94 | 103 | 5.63 | 7.57 | 9.14 |
| Entrepreneur | 1 420 | 282 | 6.37 | 9 048 | 188.79 | 112 | 6.03 | 7.14 | 9.37 |
| Gaming | 2 862 | 591 | 7.16 | 20 481 | 125.86 | 74 | 5.78 | 7.76 | 9.66 |
| Life | 3 654 | 616 | 6.30 | 23 011 | 122.55 | 70 | 5.47 | 6.86 | 8.57 |
| Movies | 3 169 | 571 | 6.24 | 19 782 | 144.04 | 87 | 5.71 | 6.88 | 8.47 |
| Politics | 2 369 | 355 | 4.09 | 9 693 | 199.36 | 108 | 5.99 | 5.37 | 6.26 |
| Science | 828 | 186 | 5.14 | 4 260 | 181.17 | 105 | 5.95 | 5.84 | 7.42 |
| Technology | 2 007 | 393 | 4.74 | 9 505 | 188.38 | 104 | 6.01 | 5.52 | 7.06 |
| Worldnews | 1 074 | 209 | 4.29 | 4 611 | 177.25 | 100 | 6.04 | 5.17 | 6.45 |

Table 2: Domain-wise dataset statistics. All average values are computed per post. **Conversation** refers to the number of conversational trees, **CtxLen** denotes the total context length of all responses under a post, **Depth** and **Width** represent the depth and maximum width of the conversation, respectively, and **UsrInt** indicates how many times the original poster replied within their own post.

*Sentiment Classification:* binary prediction of user reply polarity; (ii) *Impact Forecasting:* regression of community feedback scores (e.g., upvotes); (iii) *Personalized Text Generation:* user-specific follow-up responses in multi-turn interactions.

Following prior work (Ruddit (Hada et al., 2021), RedCaps (Desai et al., 2021), REALM (Cheng et al., 2025)), we construct user-centric trajectories from user-authored messages and metadata (reply chains, timestamps, scores). Data collection complies with platform policies and is restricted to non-commercial research. Full details appear in Appendix C.

### 3.1 CORE DEFINITIONS AND STRUCTURES

To support our three tasks in a clear formalism, we define four key components: the *message representation*, the *conversational graph*, the notion of a *conversational trajectory*, and the user's *trajectory set*. These structures unify the benchmark's multi-turn, multi-user setting and enable consistent evaluation across all tasks.

**Message Representation.** Each conversational unit is a *message* $m = (v, x, s, t)$, where $v \in \mathcal{U}$ is the author of the message, $x \in \mathcal{X}$ is the message content, $s \in \mathbb{R}$ is a numerical score indicating community feedback (e.g., upvotes), and $t \in \mathbb{R}^+$ is the timestamp. This tuple captures both the content and metadata necessary for personalization. For instance, $s$ allows us to derive sentiment (Sec. 3.2.1), while $t$ enforces a chronological ordering for multi-turn scenarios.

**Conversational Graph.** We represent the conversation space as a directed temporal graph $\mathcal{G} = (\mathcal{V}, \mathcal{E})$, where each node $m \in \mathcal{V}$ corresponds to a message $m = (v, x, s, t)$ as defined above. Each directed edge $(m_i, m_j) \in \mathcal{E}$ indicates that message $m_j$ is a reply to message $m_i$, authored by user $v_j$ and replying to user $v_i$, with the constraint that $t_j > t_i$. Formally,

$$\mathcal{G} \subseteq \left( \mathcal{V}, \ \mathcal{E} = \{(m_i, m_j) \mid t_j > t_i, \ (v_i, v_j) \in \mathcal{U}^2\} \right).$$

This makes it straightforward to follow reply chains across multiple participants, identify branching discussions, and keep track of chronological order.

**Conversational Trajectory.** A *trajectory* $\mathbf{C} = (m_1, \ldots, m_T)$ is a time-ordered path within $\mathcal{G}$, such that $(m_i, m_{i+1}) \in \mathcal{E}$ and $t_i < t_{i+1}$. It reflects a linear thread from an initial message to subsequent replies. This allow us to define context windows for tasks like follow-up generation (Sec. 3.2.3). We denote the set of all such conversational trajectories within the dataset as $\mathcal{C}$.

**User Trajectory Set.** For each user $u \in \mathcal{U}$, let $\mathcal{C}_u \subseteq \mathcal{C}$ denote all trajectories containing messages authored by $u$. This *user trajectory set* captures the user's personal conversation history, enabling personalization in tasks such as impact forecasting or sentiment classification. When the model accesses $\mathcal{C}_u$, it can incorporate user-specific context to produce more tailored outputs.

### 3.2 TASKS

We introduce each task and provide detailed data construction procedures in Appendix C.1.

### 3.2.1 TASK 1: PERSONALIZED CONVERSATIONAL SENTIMENT CLASSIFICATION

This task evaluates the ability of language models to predict the sentiment associated with a user's message during a conversation, conditioned on the conversational trajectory and the user's trajectory set. An example of this task is shown in Appendix Fig. 6a.

**Problem Setting.** Let $m_\tau \in \mathbf{C}$ be a target message authored by user $u$ (i.e., $v_\tau = u$). A binary sentiment label $y_\tau \in \{\text{positive}, \text{negative}\}$ is deterministically derived from its numerical score $s_\tau$ using a threshold $\theta$, e.g., $y_\tau = \text{positive}$ if $s_\tau > \theta$. The input consists of the conversational context $\{m_t \in \mathbf{C} \mid t_t < t_\tau\}$ and the user trajectory set $\mathcal{C}_u$. We evaluate a prompt-based LLM on predicting $y_\tau$, without training an explicit classifier. This setting focuses on testing the model's in-context ability to capture personalized interaction dynamics.

### 3.2.2 TASK 2: PERSONALIZED CONVERSATIONAL IMPACT FORECASTING

This task aims to predict the numerical score $s_\tau$ of a user's message based on prior conversation and user history, providing a finer-grained assessment than binary sentiment. An example of this task is shown in Appendix Fig. 6b.

**Problem Setting.** Given a target message $m_\tau = (v_\tau, x_\tau, s_\tau, t_\tau) \in \mathbf{C}$, where $v_\tau = u$, and given the conversational context $\{m_t \in \mathbf{C} \mid t_t < t_\tau\}$ and the user trajectory set $\mathcal{C}_u$, the goal is to predict the real-valued score $s_\tau \in \mathbb{R}$ using a prompt to elicit an estimate $\hat{s}_\tau$. Model performance is evaluated using regression metrics such as Mean Absolute Error (MAE).

This task requires more granular personalization, as it demands estimating the strength of community reception.

### 3.2.3 TASK 3: PERSONALIZED FOLLOW-UP CONVERSATIONAL TEXT GENERATION

This task focuses on generating a user's likely next response within an ongoing conversation, conditioned on the preceding conversational trajectory and the user's trajectory set. An example of this task is shown in Appendix Fig. 6c.

**Problem Setting.** Let $u \in \mathcal{U}$ be a user who participates in multi-turn discussions. For each conversational trajectory $\mathbf{C} = (m_1, \ldots, m_T) \in \mathcal{C}$, we consider a target message $m_\tau \in \mathbf{C}$ authored by user $u$ (i.e., $v_\tau = u$).

It predicts the textual content $x_\tau$ of this message, given all prior messages in the same trajectory:

$$\{m_t \in \mathbf{C} \mid t < t_\tau\}.$$

This conditioning captures the conversational flow leading up to the target message. Additionally, the user's trajectory set $\mathcal{C}_u \subseteq \mathcal{C}$, constructed from past messages authored by $u$, is provided as a source of long-term personalization signals. This leads to the generation objective:

$$p(x_\tau \mid \{m_t \in \mathbf{C} \mid t < t_\tau\}, \mathcal{C}_u).$$

We define the set of held-out generation targets for user $u$ as:

$$\mathcal{H}_u = \{m_\tau \mid \mathbf{C} \in \mathcal{C}_u, m_\tau \in \mathbf{C}, v_\tau = u\}.$$

Model performance is evaluated by comparing the generated text $\hat{x}_\tau$ with the ground-truth $x_\tau$ for all $m_\tau \in \mathcal{H}_u$.

## 3.3 EVALUATION

We adopt a unified evaluation framework with task-specific metrics for classification, regression, and generation. All evaluations follow a temporally consistent setting to reflect deployment constraints.

**Temporal Setting.** Test instances are ordered by timestamp. For each target message $m_\tau \in \mathbf{C}$, only preceding context $\{m_t \in \mathbf{C} \mid t < t_\tau\}$ and the user's past trajectories $\mathcal{C}_u$ are accessible, while $m_\tau$ itself is masked. This prevents information leakage and preserves temporal realism.

### 3.3.1 CLASSIFICATION METRICS

For sentiment classification, we report Accuracy, F1, and Matthews Correlation Coefficient (MCC) (Chicco & Jurman, 2020; Boughorbel et al., 2017; Chicco et al., 2021). Accuracy measures overall correctness, F1 balances precision and recall, and MCC remains reliable under the 1:6 class imbalance by considering all entries of the confusion matrix.

### 3.3.2 REGRESSION METRICS

For regression tasks (e.g., impact forecasting), we use Root Mean Squared Error (RMSE) and Mean Absolute Error (MAE). RMSE penalizes large deviations, while MAE is more robust to outliers (Chai et al., 2014; Hodson, 2022). Together, they capture both sensitivity and stability.

### 3.3.3 TEXT GENERATION METRICS

For personalized text generation, we report lexical metrics—ROUGE-1, ROUGE-L (Lin, 2004), BLEU (Papineni et al., 2002), METEOR (Banerjee & Lavie, 2005)—and semantic similarity via SBERT-based (Reimers & Gurevych, 2019) cosine similarity. Lexical metrics capture $n$-gram overlap and sequence structure, while SBERT reflects semantic alignment, yielding a comprehensive view of fluency and relevance.

## 4 METHODOLOGY: PROMPTING AND INFERENCE

We present a unified pipeline for evaluating LLMs on three *personalized* conversational tasks—sentiment classification, impact forecasting, and next-text generation (see Section 3.2 for details). Our pipeline consists of: (i) *Instance Construction*, where we identify test messages and relevant context for each user, (ii) *In-Context Prompt Construction*, where we build a user-specific prompt, and (iii) *LLM Inference*, where the model produces task-specific predictions.

### 4.1 INSTANCE CONSTRUCTION

We begin by selecting *test instances* from the conversational graph $\mathcal{G} = (\mathcal{V}, \mathcal{E})$. Each user $u \in \mathcal{U}$ has a set $\mathcal{H}_u$ of messages designated for evaluation. A message $m_\tau = (v_\tau, x_\tau, s_\tau, t_\tau)$ is considered a test instance if $v_\tau = u$ and belongs to some trajectory $\mathbf{C} \subseteq \mathcal{C}_u$.

**Context and User History.** For every test instance $m_\tau$, we gather: $\{ m_t \in \mathbf{C} \mid t_t < t_\tau \}$ as the *temporal context* (all messages preceding $m_\tau$ in the same trajectory). We also include the rest of the user's data $\mathcal{C}_u \setminus \mathbf{C}$ to capture cross-trajectory patterns. Depending on the task:

- **Classification & Regression:** The text $x_\tau$ is revealed the model predicts sentiment $y_\tau$ or score $s_\tau$.
- **Text Generation:** The text $x_\tau$ is hidden, and the model must generate it using only the conversation context and user's additional history.

By structuring each test instance this way, we ensure the model has a realistic snapshot of the conversation plus relevant user-specific cues, without leaking future turns.

### 4.2 IN-CONTEXT PROMPT FORMULATION

To simulate in-context learning, we include a demonstration example within the prompt. This demonstration is sampled from a different trajectory in the same graph $\mathcal{G}$, containing earlier messages authored by the same user $u$. From the current trajectory that includes $m_\tau$, we extract a prefix ending before the target message and pair it with the corresponding label or content, depending on the task.

Let $\phi(\cdot, \cdot)$ denote the prompt construction. The input prompt $P_{u,\tau}$ for message $m_\tau$ is defined as:

$$P_{u,\tau} = \phi\left(\{m_t \in \mathbf{C} \mid t_t < t_\tau\}, \mathbf{d}_u, \mathcal{C}_u \setminus \mathbf{C}\right),$$

where $\mathbf{d}_u$ is the sampled demonstration and $\mathcal{C}_u \setminus \mathbf{C}$ provides user history outside the current trajectory. We implement the in-context prompt in all tasks (Appendix D). More implementation details are shown in Appendix C.3, and prompt templates for each task are shown in Appendix D.

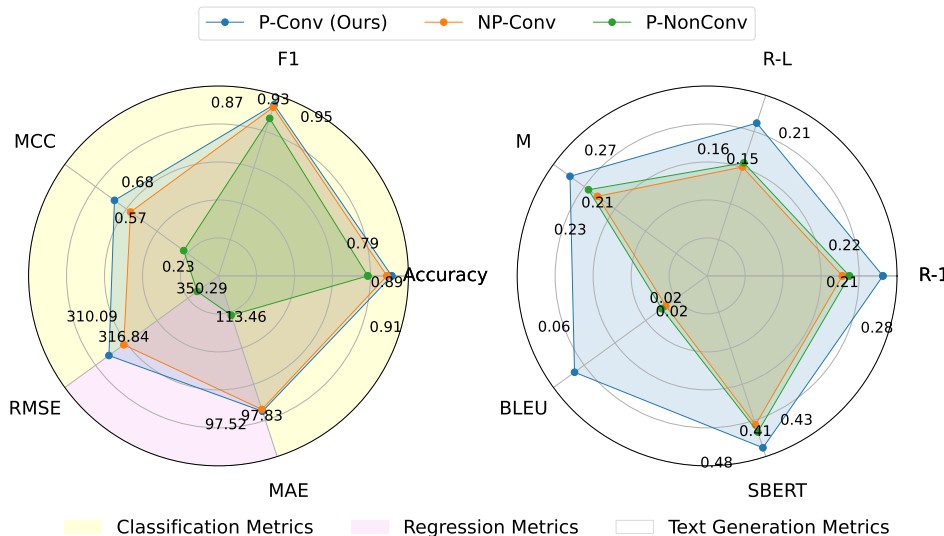

Figure 2: Performance of GPT-4.1 on our personalized conversation benchmark. Incorporating personalized conversational context significantly improves model performance across all tasks and evaluation metrics. Notably, the P-Conv variant consistently outperforms the non-personalized baselines (NP-Conv and P-NonConv) in classification, regression, and text generation metrics. Note: RMSE and MAE are normalized to $[0, 1]$ (higher is better), using the formulas: $\text{RMSE}_{\text{scaled}} = \frac{360 - \text{RMSE}}{70}$ and $\text{MAE}_{\text{scaled}} = \frac{120 - \text{MAE}}{30}$. For radar charts showing the performances of other models, please see Fig. 7- 10 in Appendix.

| Task | Metric | PERFORMANCE METRICS (**P-Conv (Ours)** \| P-NonConv) | | | | |
| | | GPT-4.1 | GPT-4o-mini | Claude-3.5 | LLaMA3.3 | DeepSeek-R1 |
|---|---|---|---|---|---|---|
| **Person. Conv. Senti. Classif.** (Sec. 3.2.1) | Accuracy ↑ | **0.9122** \| 0.7862 | **0.6875** \| 0.6562 | **0.9109** \| 0.8192 | **0.8458** \| 0.7305 | **0.8853** \| 0.7092 |
| | F1 ↑ | **0.9481** \| 0.8720 | **0.7895** \| 0.7640 | **0.9474** \| 0.8908 | **0.8401** \| 0.7495 | **0.8848** \| 0.7362 |
| | MCC ↑ | **0.6770** \| 0.2266 | **0.2268** \| 0.1870 | **0.6721** \| 0.3666 | **0.4420** \| 0.2333 | **0.6070** \| 0.2586 |
| **Person. Conv. Impact Forec.** (Sec. 3.2.2) | RMSE ↓ | **310.09** \| 350.29 | **310.23** \| 351.50 | **282.48** \| 344.75 | **319.83** \| 350.43 | **300.03** \| 353.80 |
| | MAE ↓ | **97.52** \| 113.46 | **99.64** \| 115.80 | **85.39** \| 109.27 | **101.25** \| 113.18 | **89.59** \| 112.52 |
| **Person. Conv. Next-Text Gen.** (Sec. 3.2.3) | ROUGE-1 ↑ | **0.2777** \| 0.2248 | **0.2491** \| 0.2121 | **0.2161** \| 0.1645 | **0.2055** \| 0.1540 | **0.1786** \| 0.1359 |
| | ROUGE-L ↑ | **0.2115** \| 0.1565 | **0.1906** \| 0.1470 | **0.1719** \| 0.1130 | **0.1572** \| 0.1009 | **0.1395** \| 0.0911 |
| | METEOR ↑ | **0.2677** \| 0.2316 | **0.2120** \| 0.2198 | **0.1913** \| 0.1636 | **0.1838** \| 0.1659 | **0.1649** \| 0.1401 |
| | BLEU ↑ | **0.0604** \| 0.0206 | **0.0330** \| 0.0170 | **0.0549** \| 0.0123 | **0.0480** \| 0.0089 | **0.0423** \| 0.0083 |
| | SBERT ↑ | **0.4757** \| 0.4322 | **0.4381** \| 0.3982 | **0.3942** \| 0.3512 | **0.3733** \| 0.3339 | **0.3699** \| 0.3307 |

Table 3: Main results across the three personalized conversational benchmark tasks. Within each cell, the **Personalized Conversational (P-Conv, bolded, left)** is shown separated by a vertical line from the Personalized Non-Conversational performance (P-NonConv, right) for the respective models. Results are computed over the entire dataset, which consists of data from 10 domains. For results showing performances for each individual domain, please see Table 10-19 in Appendix. We report standard metrics for classification (Accuracy, F1, MCC), regression (RMSE, MAE), and text generation (ROUGE, METEOR, BLEU, SBERT).

## 4.3 LLM INFERENCE

Let $\mathcal{M}$ denote the language model. Given the constructed prompt $P_{u,\tau}$, inference proceeds as follows: for sentiment classification, the model predicts $\hat{y}_\tau \in \{\text{positive, negative}\}$; for impact forecasting, the model predicts a scalar $\hat{s}_\tau \in \mathbb{R}$; and for next-text generation, the model generates $\hat{x}_\tau \in \mathcal{X}$.

Predictions $\hat{y}_\tau, \hat{s}_\tau, \hat{x}_\tau$ are evaluated against their corresponding ground-truth targets using metrics specified in Section 3.3.

## 5 Experiments

In this section, we present the experimental design and results for evaluating large language models on our proposed personalized conversational benchmark. We first describe the overall experimental setup, including the selection of models and prompting strategies. We then outline the design of baseline configurations, particularly focusing on how personalization and conversational context are controlled. Finally, we report the performance of various models in three core tasks: sentence classification, impact forecasting, and next-text generation, and provide detailed comparisons and analyses to highlight the effects of personalization and interaction history. Additional results, and case studies are provided in the Appendix E.

### 5.1 Experimental Setup

We evaluate a suite of commercial and open-source LMs. Commercial models include GPT-4.1 (OpenAI, 2025), GPT-4o-mini (OpenAI, 2024), and Claude-3.5-Sonnet (anthropic, 2024); open-source models include LLaMA3.3-80B-instruct (Grattafiori et al., 2024) and DeepSeek-R1 (DeepSeek-AI et al., 2025), the latter tested in inference-only mode.

All models are run zero-shot without fine-tuning (optional fine-tuning is discussed in Appendix C.4). Prompts are manually designed per task under a unified in-context learning strategy (Sec. 4.2) with consistent formatting across models. Outputs are parsed as categorical labels or scalar values for classification and regression, and generated with greedy decoding for text generation. Evaluation covers all 10 conversational domains, with results reported as sample-level averages. Additional details are provided in Appendix C.3.

#### 5.1.1 Baselines

To demonstrate the effectiveness and utility of our benchmark, we investigate the following baselines:

**Personalized Non-conversational (P-NonConv)**: This baseline ignores conversational context and does not incorporate any prior user interactions. For each target message $m_\tau$, the input includes only the root post from the same trajectory—i.e., the first message authored by the user that initiated the conversation. In-context demonstrations are fixed across all test instances and are used solely to indicate the task format (e.g., input/output structure); they are not semantically related to the test instance. Despite the lack of conversational structure, this setting retains personalization through the use of user-authored content. It is conceptually similar to LaMP (Salemi et al., 2024) or LongLaMP (Kumar et al., 2024), depending on the generation length, as those models also operate without leveraging conversational context. *Compared to our method, P-NonConv uses only the root post and cannot adapt to later turns, while ours also conditions on the full dialogue.*

**Non-personalized Conversational (NP-Conv)**: This baseline includes conversational context, but without any personalization. For each target message $m_\tau$, the in-context examples are constructed from all messages in a randomly sampled trajectory from the dataset, regardless of whether the target user u appears in it. This provides a full conversational thread as context, but entirely unrelated to the target instance. The purpose of this baseline is to isolate and assess the effect of general conversational structure, independent of user identity or history, serving as a contrast to personalized conversation-based prompts. *Compared to our method, NP-Conv gives dialogue flow from other users but no signals from the target user, while ours aligns with both user style and context.*

Experimental details of these two baselines are shared in the Appendix. C.3.

### 5.2 Results

**Conversation history helps models understand more deeply and consistently perform better.** Experimental data clearly shows that for any task or model, adding previous conversational content improves the model's performance (see the overall comparison in Tab. 3 and in Fig. 2). For example, in the task of sentiment classification, when the GPT-4.1 model can see the conversation history, its MCC metric jumps from 0.2266 to 0.6770 (data from Tab. 3). This continuous improvement shows that if user interactions are viewed as evolving processes rather than isolated Q&A, large language models can more accurately grasp user intent, track changing topics, and understand

what was discussed previously. This ability to base understanding of the current situation on previous conversation content allows the model to move beyond simple text pattern matching to a more coherent interpretation of user needs.

**Conversational History Enhances Understanding and Reasoning.** Experimental results (Tab. 3) show that incorporating conversational history consistently improves performance across tasks and models. By treating interactions as evolving dialogues rather than isolated Q&A, LLMs better capture user intent, topic shifts, and contextual nuances crucial for reasoning. In sentiment classification, history enables detection of subtleties like sarcasm or context-dependent emotions, reflected in GPT-4.1's MCC rising from 0.2266 to 0.6770. For impact forecasting, prior turns provide baselines, as seen in Claude-3.5's MAE dropping from 109.27 to 85.39. Thus, leveraging dialogue history allows models to move beyond surface text matching toward more accurate interpretation of user needs.

**The limits of forecasting tasks show that conversation content alone is sometimes not enough.** Although conversational information helps models perform better on prediction tasks (like impact forecasting), error rates are still not low (for instance, Tab. 3 shows Claude-3.5's RMSE is still 282.48 even with conversation history). This indicates that these types of tasks are inherently complex. To estimate numerical feedback like community scores, it seems one must look beyond what's said in the conversation to some hidden factors outside the conversation, such as gradual changes in community voting habits, user reputation, or broad topic trends that are not explicitly stated in the conversation. This finding highlights a point: even though conversation history is very useful, for tasks that require a deep understanding of external situations or group dynamics, its role may be limited without integrating additional information sources or broader knowledge.

**Irrelevant conversation history can be harmful, highlighting the importance of personalization.** As shown in appendix Tab. 8, if a model is given conversation history unrelated to the current user (NP-Conv), its performance can actually worsen, sometimes even falling below the baseline of having no conversation history but knowing the user (P-NonConv). For example, in the task of generating next text, when the Claude-3.5 model was given irrelevant conversation history (NP-Conv), its SBERT metric significantly dropped from 0.3512 (for the user-specific no-conversation mode, P-NonConv) to 0.2676. This clearly shows that just providing any conversation history is not helpful; the history must be relevant and specific to the current user. Confusing conversation history introduces noise and can potentially lead the model to misunderstand the user's current meaning or the progress of the conversation. This strongly proves that truly effective conversational AI needs reliable methods to retrieve and use user-specific conversation history, making personalization a critical component, not an optional add-on.

## 6  CONCLUSION

We introduce PERSONACONVBENCH, the first benchmark for evaluating large language models on *personalized* conversational tasks involving classification, regression, and generative response. By incorporating both user history and conversational context, our setup better reflects the real-world need for models that adapt to individual user behaviors and discourse patterns. Empirical results show that leveraging personalized conversation histories yields notable performance gains across a broad range of tasks and domains. We hope this benchmark will support new research on user-centric dialogue modeling, ultimately leading to more context-aware and effective language technologies. interact with AI-driven systems in daily life.

**Limitations.** Our benchmark draws from a specific set of online discussion domains, and some tasks may not generalize to highly specialized or low-resource settings. Also, we focus primarily on English data, leaving cross-lingual personalization and cultural adaptation for future exploration.

**Impact Statement.** Our publicly available dataset and framework offer a powerful resource for building more responsive and user-focused conversational applications. We believe this will support new innovations in areas such as virtual assistants, customer support, and collaborative platforms, ultimately improving how people interact with AI-driven systems in daily life.

## 7 ETHICS STATEMENT & REPRODUCIBILITY STATEMENT

This work adheres to standard academic research practices. All data used are publicly available, and the study is intended solely for scientific and educational purposes. We do not foresee any ethical concerns arising from the content or methodology presented. For reproducibility, we have included sufficient technical details in the paper to allow other researchers to replicate our experiments. The dataset statistics, task definitions, and evaluation protocols are described in detail, and we aim to facilitate further exploration and extension by the community.

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

## A  RELATED WORK

### A.1  PERSONALIZATION IN LLMS

LaMP (Salemi et al., 2024) introduces a suite of classification and generation tasks (e.g., headline generation), each incorporating user profiles to simulate personalized language modeling scenarios. Building on this, LongLaMP (Kumar et al., 2024) extends the benchmark to long-form generation tasks such as personalized email generation and review writing. It evaluates model performance under two settings: user-based splits (to assess generalization to unseen users) and temporal splits (to track the evolution of user preferences over time). PersoBench (Afzoon et al., 2024) further benchmarks the ability of LLMs to generate persona-consistent responses, with evaluations across multiple axes such as fluency, coherence, and diversity. Given the difficulty of collecting personalization datasets at scale, Persona-SQ (Lin et al., 2024) proposes a scalable framework for generating suggested questions tailored to user backgrounds and reading goals. While prior work primarily focuses on evaluation and generation, PersonalLLM (Zollo et al., 2025) introduces an alignment dataset that captures diverse user preferences without relying on predefined personas, thereby improving the generalizability of personalized LLMs.

Recent studies have explored the personalization of LLMs to align their interactions and recommendations with individual user preferences (Chen et al., 2024). Techniques such as personalized parameter-efficient fine-tuning (PEFT) and low-rank reward modeling have been proposed to efficiently adapt LLMs to user-specific behavior patterns and preferences. For example, One PEFT Per User (OPPU) employs personal PEFT parameters to store user-specific behavior patterns (Tan et al., 2024), while LoRe leverages low-rank preference modeling to learn and generalize user-specific reward functions (Bose et al., 2025). Other works have focused on developing frameworks for personalized text generation, such as a multistage and multitask approach inspired by writing education (Li et al., 2023), and conversational health agents that utilize LLMs to provide personalized healthcare services (Abbasian et al., 2023). Additionally, studies have investigated the evaluation of LLMs' personalization capabilities, including the development of benchmarks like LaMP and metrics like EGISES to measure degree-of-personalization (Patel et al., 2024). These advances highlight the growing interest in personalizing LLMs and demonstrate the potential for improved user satisfaction and alignment.

### A.2  CONVERSATIONS

Recent works introduce benchmarks to evaluate the capabilities of LLMs in multi-turn conversations, typically without considering user-specific intent or personalization. For example, (Kwan et al., 2024; Zhao et al., 2023) focus on evaluating interaction patterns such as recollection, expansion, refinement, and follow-up. Similarly, (Mendonça et al., 2024) assess aspects like commonsense knowledge and coherence, while (Zhang et al., 2023) extend such evaluations to multilingual scenarios. Further, (Castillo-Bolado et al., 2024) explore long-term memory and continual learning in more dynamic, multi-round conversational contexts. While these benchmarks advance general-purpose multi-turn evaluation, recent studies highlight the growing importance of simulating personalized conversational trajectories, especially for users with extremely sparse feedback (Ashok & May, 2024). For users with only a few short conversations—generating plausible user-specific dialogue becomes critical for personalization, learning, and evaluation. This is further motivated by the increasing reliance on simulation-based data generation, as large models approach performance plateaus due to finite high-quality human data (Villalobos et al., 2024; Zhang et al., 2024). Simulation enables novel, interactive data generation beyond existing corpora, supporting better alignment and robustness especially in the interactions between modalities (Zhang et al., 2025; Zhao et al., 2025). However, synthetic data also introduces risks such as model collapse and degraded generalization if not properly managed (Shumailov et al., 2024). These insights underscore the need for personalized and simulation-aware evaluation frameworks in future conversational benchmarks.

## B  APPLICATIONS ENABLED BY PERSONACONVBENCH

PERSONACONVBENCH enables several real-world applications where large language models must reason over user-specific histories and multi-turn conversational context. We highlight four representative use cases:

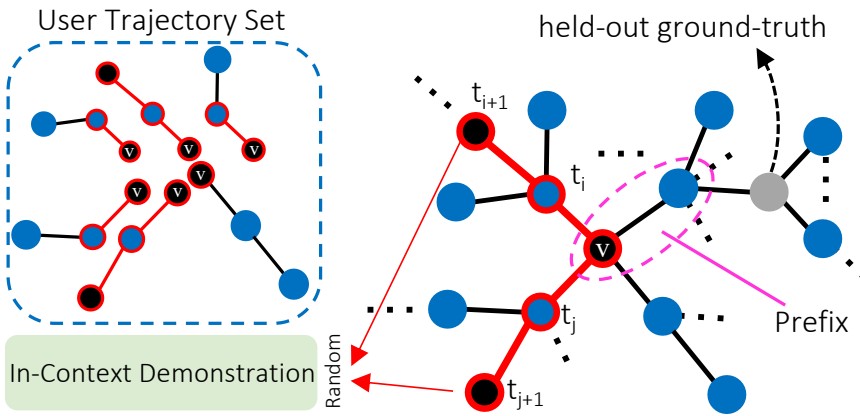

Figure 3: In-context prompt construction for personalized conversational inference. Given a held-out user trajectory set, we sample a prefix from the current test thread and draw a demonstration example from a different random thread by the same user. The prefix and demonstration, along with user history outside the test thread, are composed into a single prompt for in-context learning. This unified formulation supports three tasks—sentiment classification, impact forecasting, and next-text generation—by conditioning the LLM on personalized multi-turn context without future leakage.

**Personalized Customer Support.** In customer service platforms (Patel & Trivedi, 2020), users often return with follow-up questions or ongoing issues. PERSONACONVBENCH enables the development of LLM-based agents that recall prior interactions, model user sentiment trends, and tailor responses accordingly. This leads to more consistent and empathetic support across sessions.

**Adaptive Virtual Assistants.** Virtual assistants must adjust their responses based on how individual users interact over time (Lamontagne et al., 2014). Using structured user trajectories from PERSONACONVBENCH, assistants can be trained to personalize reminders, recommendations, and clarification strategies, improving long-term usability and trust.

**Mental Health and Social Support.** Conversational agents for wellness and peer support must track emotional dynamics across multiple conversations (Tutun et al., 2023; Li et al., 2024). PERSONACONVBENCH allows models to detect user-specific sentiment shifts and generate context-sensitive, supportive responses—important for early intervention and engagement in sensitive settings.

**Education and Tutoring Systems.** Personalized tutoring systems can leverage student interaction history to adapt explanations, adjust difficulty, and identify knowledge gaps (Lin et al., 2023). The benchmark's multi-domain conversations and task diversity provide a foundation for building LLM-based tutors that reason over prior dialogue turns and tailor feedback to individual learning trajectories.

These applications demonstrate how PERSONACONVBENCH facilitates the development of user-aware, context-sensitive LLMs that perform reliably in complex, real-world interaction settings.

## C ADDITIONAL BENCHMARK DETAILS

### C.1 DATA CONSTRUCTION

We introduce the detailed approach used to construct the data for each task. Our initial information was obtained from publicly available discussions on an online community platform. An automated system was used to review content from a specific set of communities. Within these communities, the system examined popular discussions in a sequential manner until a sufficient volume of information was gathered.

The collection method was designed as follows: For any given discussion, we identified the author. If the author was new to our collection, they were added to a designated group of users, and we would then review the public activity histories of these new users. For each user in our group, we examined the discussions they started to see if they met several specific conditions:

- The discussion had to be within our chosen communities.
- More than $N_u = 4$ unique individuals had to be involved in the conversation.
- The original author needed to have contributed at least $N_r = 4$ responses within the comment threads.
- The author's contributions had to span at least two separate sub-threads of the discussion.

When a discussion initiated by a user met all these requirements, it was marked as a potential candidate and associated with that user. After reviewing all of a user's initiated discussions, we checked if they had accumulated at least $N_p = 3$ such candidate discussions. If this threshold was met, the relevant information from those discussions was saved as a structured data file for the subsequent phases of our research.

All information gathered for this study was sourced from public posts on Reddit and is consistent with the platform's user agreement.

**Personalized Conversational Sentiment Classification.** For the Personalized Conversational Sentiment Classification task, we implemented specific post-screening mechanisms and prompt design techniques. To better reflect the model's semantic understanding capabilities, we favored comments with stronger influence, as indicated by our requirement for the absolute value of the score of candidate replies. Furthermore, according to our statistics, the ratio of positive to negative scores for the single top-level comment or reply with the highest absolute score in each post of the original data was approximately 11:1. This led to a severe class imbalance in the binary distribution, allowing models that tended to classify a large volume of content as positive to achieve deceptively better accuracy. To address these issues, we designed the following screening mechanism for conversational data:

- Author replies with the highest absolute score (not less than $N_s = 3$) and whose replied-to object was not `[deleted]` were designated as prediction targets. Posts that did not meet this condition were skipped.
- Author replies with the second-highest absolute score (not less than $N_{s2} = 2$) and whose replied-to object was not `[deleted]` were designated as few-shot demonstration targets. Posts failing this condition were also skipped.
- After screening, posts whose prediction-target replies had a positive score but ranked in the bottom 55% by absolute score were further filtered out.

This process ultimately retained approximately 6,000 posts, wherein both negative and positive sentiments possessed strong emotional coloring or influence, and were in a more reasonable ratio of 5:1. In prompt design, it was necessary to retain all contextual information within the current post except for the information to be predicted. Simultaneously, the model needed to understand the tree-node-like relationships between contextual elements. To achieve this, we assigned an ID to each post/comment/reply, using the format `{parent-node-id}-{current-node-child-id}` as the complete ID for a given node, and provided an explanation of this encoding scheme to enable the model to comprehend the interrelations within the context. Additionally, we used few-shot examples that closely mirrored the actual task pattern to explicitly instruct the model to respond using our required reply template, thereby standardizing the model's output.

**Personalized Conversational Impact Forecasting.** For Personalized Conversational Impact Forecasting, the screening mechanism was identical to that of the previous task, with alterations only in the task description and the stored true label, which was changed from positive/negative to a specific numerical value.

**Personalized Follow-up Conversational Text Generation.** For Personalized Follow-up Conversational Text Generation, we directly adopted the author's most recent reply within their own post as the candidate prediction reply, unless the current reply was '[deleted]' or its length was excessively

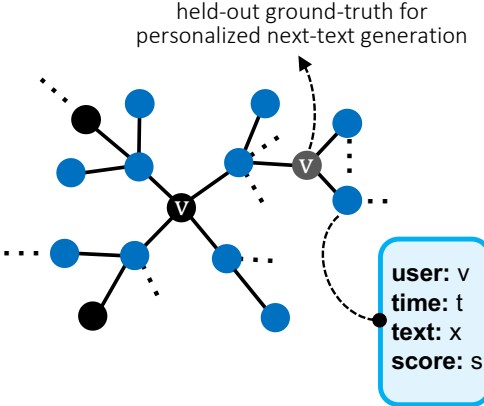

Figure 4: Illustration of the personalized conversational setting in PERSONACONVBENCH. The center black node represents a user $v$ who initiates a post, leading to multiple conversational trajectories as other users respond. Over time, user $v$ replies to some of these users, forming deeper branches in the graph. For each reply made by user $v$, we can construct a prediction task—either classifying the response's sentiment, forecasting its community score, or generating the response text—based on the earlier parts of the same trajectory and additional user-specific history. These tasks rely on realistic multi-user, multi-turn settings with graph-structured conversational data. Each message is annotated with username $v$, timestamp $t$, message content $x$, and feedback score $s$, supporting fine-grained personalization across all task types.

long or short. If the latter occurred, we proceeded sequentially in chronological order. If, after such an adjustment, the context size fell below expectations, the current post was skipped. The context input specifications were consistent with the previous two tasks. However, the screening mechanism was comparatively more lenient, no longer imposing stringent requirements on influence.

## C.2 TASK SCHEMA

As shown in Figure 6, we illustrate representative prompt formats for the three core personalized tasks in PERSONACONVBENCH: sentiment classification, impact forecasting, and follow-up text generation. Each prompt is constructed by combining two major components: (i) a full conversational trajectory, including all messages leading up to the target response, and (ii) the user trajectory set.

In each case, the input prompt includes a natural language instruction followed by structured conversation data, where the prediction target is masked. For the classification task (Fig. 6a), the model is asked to classify the sentiment (positive or negative) of a specific reply. For regression (Fig. 6b), the task is to predict the exact community score associated with the target reply. For generation (Fig. 6c), the model is expected to produce the content of the user's response. The associated user profile remains constant throughout each prompt and includes the full conversation tree rooted at the user's original post.

These examples highlight the unified input structure across tasks, enabling evaluation of language models under a consistent personalization and in-context learning framework.

## C.3 EXPERIMENTAL DETAILS

As shown in Fig. 3, we showcase the in-context prompt formulation process. For the Personalized Conversational Sentiment Classification and Personalized Conversational Impact Forecasting experiments, we used a temperature of 0 and a maximum output length of 15 tokens. DeepSeek-R1 was the exception—it was forced to use a maximum of 1200 tokens in order to provide a sufficiently long *[think]* span.

For the Personalized Follow-up Conversational Text Generation experiment, we set the temperature to 0.7 and the maximum output length to 256 tokens. All models except Claude were configured to generate 10 candidate responses; Claude imposed restrictions on non-primary outputs, so it was lim-

```
INPUT:
### FEW-SHOT EXAMPLE ###
Consider the following dialogue tree context (the author's t
arget reply is missing, and subsequent replies in that speci
fic branch are truncated):
[
  {
    "author": "USeR...",
    "body": "If this was me I ...",
    "score": 2,
    "timestamp": "",
    "id": "ulp3-c13",
    "parent_id": "ulp3",
    "type": "comment",
    "replies": [...]
  }
]
The author of the post, 'USER...', made a reply to a comment
 by 'USeR...' (ID: ulp3-c13) which says: "If this was me I..
.". (This reply by 'USER...' had ID: ulp3-c13-c14).
Predicted body text for USER...'s reply:...
### ACTUAL TASK ###
Now, given the following full post conversation context (the
 author's target reply is missing):
{
  "title": "Will paying off half ...",
  "author": "USER...",
  "url": "...",
  "score": 5,
  "num_comments": 18,
  "content": "Husband and I are 73 and 69...",
  "timestamp": "2021-05-19 04:25:36",
  "comments": [ …full thread as before… ],
  "__sub__": "entrepreneur",
  "__srcfile__": "Art...",
  "id": "ulp3",
  "type": "post"
  "comments": [
    {
```

(a) Prompt construction and input context.

```
    "id": "ulp3-c1",
    "parent_id": "ulp3",
    "author": "USEr-...",
    "body": "Back up for a sec:...",
    "score": 40,
    "timestamp": "...",
    "type": "comment",
    "replies": [
      {...
}
The author of the post, 'USER...', made a reply to a comment
 by 'eru...' (ID: ulp3-c1-c4-c5) which says: "One note: once
 you pay off the cards, I wouldn't necessarily close them, s
ince that can hurt your credit score by lowering the number
of active accounts, lowering your total credit limit, and po
tentially lowering the average age of accounts. Unless you h
ave a good reason to close a specific card (like an annual f
ee), I'd leave it open and just not use it.". (This reply by
 'USER...' has ID: ulp3-c1-c4-c5-c6).
Predict the body text of this reply by 'USER...'. Respond ON
LY with the predicted text content of the body.
--------------------------------------------
OUTPUT (After Postprocessing):
"best_sbert_candidate_text": "Thank you, that's helpful to k
now! We weren't planning to close any cards, but I wasn't su
re if leaving them open was the right thing. We'll keep them
 open and just try not to use them. Appreciate your advice!
"
--------------------------------------------
EVAL:
"true_label": "I wasn't referring to closing them, just not
use them. I'd prefer to use only 1 or 2 cards, such as Disco
ver.",
"SBERT": 0.7201
"BLEU": 0.0192
"ROUGE-1": 0.3582
"ROUGE-L": 0.2388
"METEOR": 0.3496
```

(b) Generated output, true label, and evaluation metrics.

Figure 5: Case study of the Personalized Conversational Follow-up Text Generation task. The model is asked to generate a masked reply based on the full conversational context and a user-specific example. Output is evaluated using both lexical (e.g., ROUGE, BLEU) and semantic (e.g., SBERT) metrics.

```
"input": "...Predict whether the
score for User_ID_11451's reply (
ID: 'u1p1-c3-c4') will be positiv
e or negative. Respond with only
'positive' or 'negative'...",
"output": "positive",
-------------------------------
"profile": "
{
  "post_data_store": {
      "title": "...",
      "author": "...",
      "url": "...",
      "score": ...,
      "num_comments": ...,
      "content": "...",
      "timestamp": "...",
      "__sub__": "...",
      "id": "...",
      "type": "post",
      "comments": [
        {
         "author": "...",
         "body": "...",
         "score": ...,
         "timestamp": "...",
         "id": "...",
         "parent_id": "...",
         "type": "comment",
         "replies": [
           { /* ... */ }
         ]
        }
      ]
  }
}"
```
(a) Sentiment classification.

```
"input": "...Predict the exact sc
ore of User_ID_11451's reply (ID:
'u1p1-c3-c4'). Respond with only
the exact number...",
"output": "114514",
-------------------------------
"profile": "
{
  "post_data_store": {
      "title": "...",
      "author": "...",
      "url": "...",
      "score": ...,
      "num_comments": ...,
      "content": "...",
      "timestamp": "...",
      "__sub__": "...",
      "id": "...",
      "type": "post",
      "comments": [
        {
         "author": "...",
         "body": "...",
         "score": ...,
         "timestamp": "...",
         "id": "...",
         "parent_id": "...",
         "type": "comment",
         "replies": [
           { /* ... */ }
         ]
        }
      ]
  }
}"
```
(b) Impact Forecasting.

```
"input": "...Predict the text bod
y for User_ID_11451's reply (ID:
'u1p1-c3-c4'). Respond with only
body content...",
"output": "Yaju Senpai and Billy
are best internet couple I've eve
n seen XDDD!",
-------------------------------
"profile": "
{
  "post_data_store": {
      "title": "...",
      "author": "...",
      "url": "...",
      "score": ...,
      "num_comments": ...,
      "content": "...",
      "timestamp": "...",
      "__sub__": "...",
      "id": "...",
      "type": "post",
      "comments": [
        {
         "author": "...",
         "body": "...",
         "score": ...,
         "timestamp": "...",
         "id": "...",
         "parent_id": "...",
         "type": "comment",
         "replies": [
           { /* ... */ }
         ]
        }
      ]
  }
}"
```
(c) Follow-up Text Generation

Figure 6: Task schema for the three personalized tasks. The target field is masked in each case. All prompts include the user profile and full conversational context leading up to the target.

ited to a single candidate to preserve performance. The final score for each input was computed by selecting, from those 10 candidates, the one with the highest SBERT score as the "Best Response."

For the baseline experiment prompts, compared to the full Conversational Prompt, we removed the complete context and retained only a limited portion of the dialogue to make the task solvable. The experimental details remain identical to those in the Conversational setting.

### C.4 OPTIONAL FINE-TUNING EXTENSION

While our primary focus is on in-context prompting, we outline a possible fine-tuning extension for settings where labeled user-specific data is available. In this setup, the base model $\mathcal{M}$ can be adapted using supervised training over constructed input-output pairs $\{(P_{u,\tau}^{\text{train}}, y_\tau)\}$, where $P_{u,\tau}^{\text{train}}$ is the context-aware prompt and $y_\tau$ is the task-specific target.

Fine-tuning proceeds by minimizing a standard objective:

$$\mathcal{L}_{\text{FT}} = \sum_\tau \ell \left( \mathcal{M}(P_{u,\tau}^{\text{train}}), \, y_\tau \right),$$

where $\ell(\cdot, \cdot)$ denotes the appropriate task-specific loss function (e.g., cross-entropy, regression loss, or generation loss).

This extension allows the model to learn personalized behaviors directly from training data. We leave its empirical study to future work, as our main experiments focus on zero-shot inference via prompting.

### D PROMPT DEMONSTRATION

The complete prompts for all three tasks are provided in Tables 4 to 7.

### FEW-SHOT EXAMPLE ###
Referenced Post (ID: u1p1):
- Subreddit: r/entrepreneur
- Author: u/User_ID_81778
- Score: 89
- Timestamp: 2024-07-21T23:16:43Z
- Title: "Please explain 401k to senior with new job"
- Content Summary: "I'm 72 and just started a new job in April...{*more content is omitted*}" (Full post content and dialogue tree are available in the data store under post reference ID 'u1p1')
The author of the post, 'User_ID_81778', made a reply.
Parent Comment (that 'User_ID_81778' replied to, ID: 'u1p1-c3-c4-c5'): Author: 'User_ID_19191', Body Summary: "RMDs out of 401ks are not...{*more content is omitted*}"
Author's Reply (ID: 'u1p1-c3-c4-c5-c6'): Timestamp: '2024-07-22T03:09:14Z', Body: "Ah good, thanks for that info."
(The actual score of this Author's Reply 'u1p1-c3-c4-c5-c6' is hidden. Its full context is in post ID 'u1p1' in the data store.)

Based on all this information, predict if the score for User_ID_81778's reply (ID: 'u1p1-c3-c4-c5-c6') was positive or negative.
Predicted score sentiment:
positive

- - - - - - - - - - - - - - - - - - - - - - - - - - - - - - - - - - - - - - - - - - - - - - - - - - - - - - -

### ACTUAL TASK ###
Referenced Post (ID: u1p1):
- Subreddit: r/entrepreneur
- Author: u/User_ID_81778
- Score: 89
- Timestamp: 2024-07-21T23:16:43Z
- Title: "Please explain 401k to senior with new job"
- Content Summary: "I'm 72 and just started a new job in April..." (Full post content and dialogue tree are available in the data store under post reference ID 'u1p1')
The author of the post, 'User_ID_81778', made another reply (or the primary reply we are interested in).
Parent Comment (that 'User_ID_81778' replied to, ID: 'u1p1-c3'): Author: 'User_ID_11451', Body Summary: "¿What is profit sharing...{*more content is omitted*}"
Author's Reply (ID: 'u1p1-c3-c4'): Timestamp: '2024-07-21T23:52:30Z', Body: "Thanks...{*more content is omitted*}"
(Your task is to predict the score sentiment of this Author's Reply 'u1p1-c3-c4'. Its full context is in post ID 'u1p1' in the data store.)

Predict whether the score for User_ID_81778's reply (ID: 'u1p1-c3-c4') will be positive or negative. Respond with only 'positive' or 'negative'.
Predicted score sentiment:

Table 4: LLM prompt template for personalized conversational sentiment classification.

### FEW-SHOT EXAMPLE (With Full Conversation Context) ###
Referenced Post (ID: N/A):
- Subreddit: r/entrepreneur
- Author: u/User_ID_81778
- Score: 89
- Timestamp: 2024-07-21T23:16:43Z
- Title: "Please explain 401k to senior with new job"
- Content Summary: "I'm 72 and just started a new job in April...{*more content is omitted*}" (Full post content and dialogue tree are available in the data store under post reference ID 'N/A')
The author of the post, 'User_ID_81778', made a reply.
Parent Comment (that 'User_ID_81778' replied to, ID: 'u1p1-c3-c4-c5'): Author: 'User_ID_81342', Body Summary: "RMDs out of 401ks are not...{*more content is omitted*}"
Author's Reply (ID: 'u1p1-c3-c4-c5-c6'): Timestamp: '2024-07-22T03:09:14Z', Body: "Ah good, thanks for that info."
(The actual score of this Author's Reply 'u1p1-c3-c4-c5-c6' is hidden. Its full context is in post ID 'u1p1' in the data store.)

Predict the specific numerical score for 'User_ID_81778's reply (ID: 'u1p1-c3-c4-c5-c6').
RULES FOR YOUR RESPONSE (for this example and actual task):
- Internally decide if the score is positive or negative (do not state this step).
- The score's absolute value MUST be 3 or greater.
- Output ONLY the integer. NOTHING ELSE. For example: -7 or 5 or 25.

Score:
10

- - - - - - - - - - - - - - - - - - - - - - - - - - - - - - - - - - - - - - - - - - - -

### ACTUAL TASK (With Full Conversation Context) ###
Referenced Post (ID: N/A):
- Subreddit: r/entrepreneur
- Author: u/User_ID_81778
- Score: 89
- Timestamp: 2024-07-21T23:16:43Z
- Title: "Please explain 401k to senior with new job"
- Content Summary: "I'm 72 and just started a new job in April...{*more content is omitted*}"(Full post content and dialogue tree are available in the data store under post reference ID 'N/A')
The author of the post, 'User_ID_81778', made another reply (or the primary reply we are interested in).
Parent Comment (that 'User_ID_81778' replied to, ID: 'u1p1-c3'): Author: 'User_ID_11451', Body Summary: "¿What is profit sharing...{*more content is omitted*}"
Author's Reply (ID: 'u1p1-c3-c4'): Timestamp: '2024-07-21T23:52:30Z', Body: "Thanks...{*more content is omitted*}"
(Your task is to predict the specific numerical score of this Author's Reply 'u1p1-c3-c4'. Its score is hidden as '[SCORE_TO_PREDICT]' in its full context within post ID 'u1p1' in the data store.) (Hint: the score is expected to be positive).

Predict the specific numerical score for 'User_ID_81778's reply (ID: 'u1p1-c3-c4').
RULES FOR YOUR RESPONSE:
- Internally decide if the score is positive or negative (do not state this step).
- The score's absolute value MUST be 3 or greater.
- Output ONLY the integer. NOTHING ELSE. For example: -7 or 5 or 25.

Score:

Table 5: LLM prompt template for personalized conversational impact forecasting.

### FEW-SHOT EXAMPLE ###
Post Information:
- Subreddit: r/entrepreneur
- Post ID: u1p1
- Author: u/User_ID_81778
- Score: 89
- Timestamp: 2024-07-21T23:16:43Z
- Title: "Please explain 401k to senior with new job"
- Content: "I'm 72 and just started a new job in April...*{more content is omitted}*"
The author, 'User_ID_81778', made a reply in this post.
Parent Comment (that 'User_ID_81778' replied to): Author: 'gohblu', ID: 'u1p1-c3-c4-c5',
Body: "RMDs out of 401ks are not...*{more content is omitted}*"
Author's Reply Details: ID: 'u1p1-c3-c4-c5-c6', Timestamp: '2024-07-22T03:09:14Z',
Body: "Ah good, thanks for that info."
(The actual score of this reply is hidden for this prediction task).
Dialogue Tree Context (the score for '{main_author_name}'s reply ID
'{main_target_reply_id}' is hidden):
{
    "title": "Please explain 401k to senior with new job",
    "author": "User_ID_81778",
    "url": "https://www.reddit.com...*{more content is omitted}*",
    "score": 89,
    "num_comments": 36,
    "content": "I'm 72 and just started a new job in April...*{more content is omitted}*"
    "timestamp": "2024-07-21 23:16:43",
    "comments": [
    {
        "author": "User_ID_81964",
        "body": "General rule is...*{more content is omitted}*",
        "score": 30,
        "timestamp": "2024-07-22 09:00:57",
        "replies": [
        {
            "author": "User_ID_81778",
            "body": "No other investments.",
            "score": 3,
            "timestamp": "2024-07-22 13:49:55",
            "replies": [],
            "id": "u1p1-c1-c2",
            "parent_id": "u1p1-c1",
            "type": "comment"
        }
        ],
        "id": "u1p1-c1",
        "parent_id": "u1p1",
        "type": "comment"
    },
    { ... remaining comment tree unchanged ... }
    ],
    "__sub__": "entrepreneur"
    "id": "u1p1",
    "type": "post"
}
Based on all this information, predict if the score for User_ID_81778's reply (ID: 'u1p1-c3-c4-c5-c6') was positive or negative.
Predicted score sentiment:
positive

Table 6: LLM prompt template for personalized follow-up conversational text generation (part 1).

### ACTUAL TASK ###
Post Information:
- Subreddit: r/entrepreneur
- Post ID: u1p1
- Author: u/User_ID_10388
- Score: 89
- Timestamp: 2024-07-21T23:16:43Z
- Title: "Please explain 401k to senior with new job"
- Content: "I'm 72 and just started a new job in April...{*more content is omitted*}"
The author, 'User_ID_10388', made another reply in this post (or the primary reply we are interested in).
Parent Comment (that 'User_ID_10388' replied to): Author: 'User_ID_10492', ID: 'u1p1-c3', Body: "¿What is profit sharing...{*more content is omitted*}"
Author's Reply Details: ID: 'u1p1-c3-c4', Timestamp: '2024-07-21T23:52:30Z', Body: "Thanks...{*more content is omitted*}"
(Your task is to predict the score sentiment of this reply).
Dialogue Tree Context (the score for '{main_author_name}'s reply ID '{main_target_reply_id}' is hidden):
{ ... identical JSON structure for the second part ... }

Predict whether the score for User_ID_10388's reply (ID: 'u1p1-c3-c4') will be positive or negative. Respond with only 'positive' or 'negative'.
Predicted score sentiment:

Table 7: LLM prompt template for personalized follow-up conversational text generation (part 2).

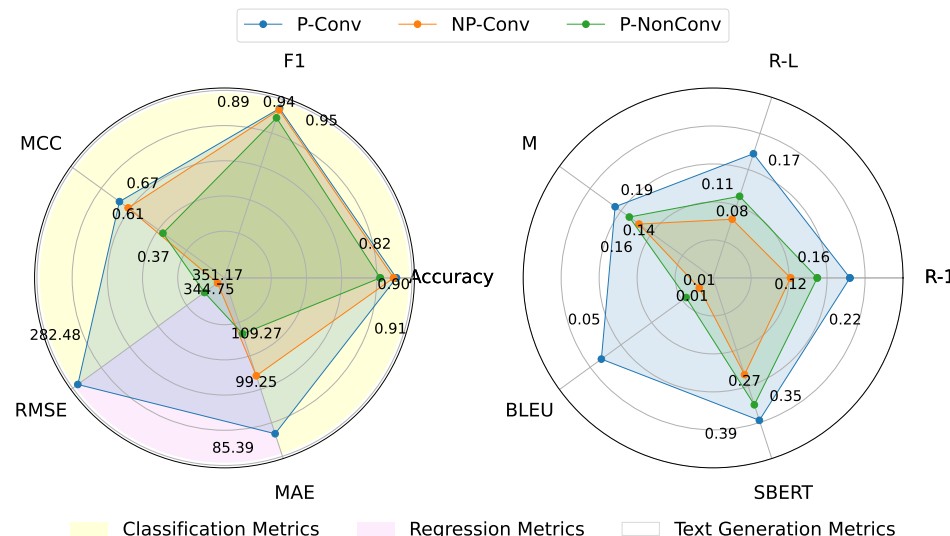

Figure 7: Performance of `Claude-3.5-Sonet` on our personalized conversation benchmark (left radar chart for classification and regression; right one for generation).

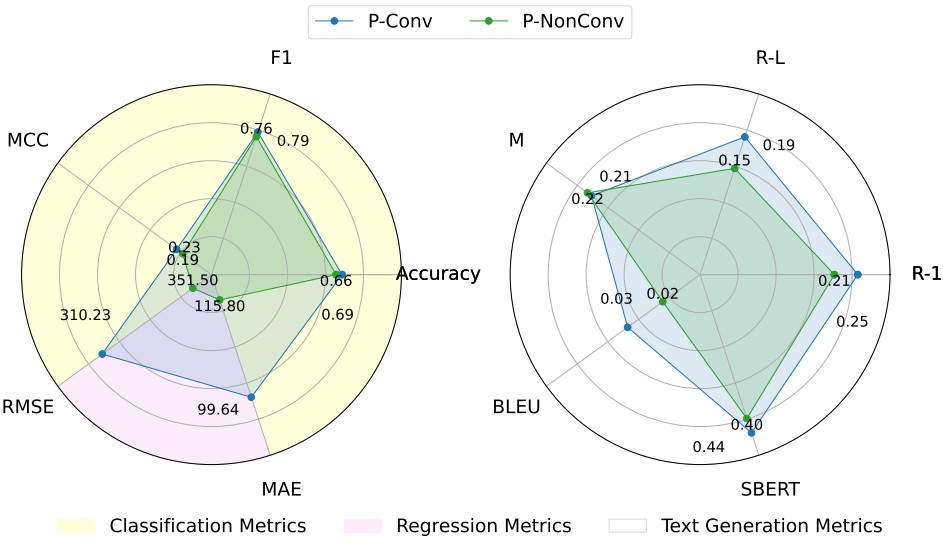

Figure 8: Performance of `GPT-4o-mini` on our personalized conversation benchmark (left radar chart for classification and regression; right one for generation).

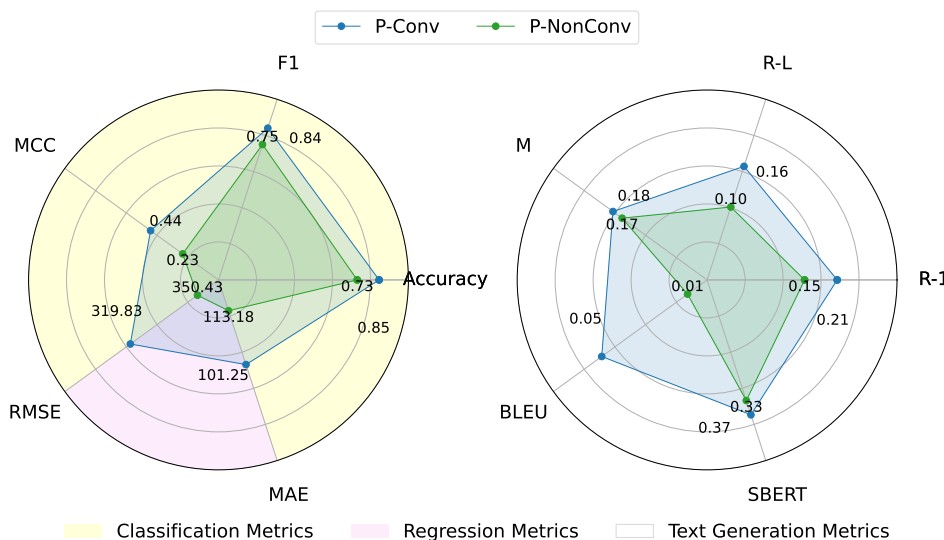

Figure 9: Performance of `LLaMA3.3` on our personalized conversation benchmark (left radar chart for classification and regression; right one for generation).

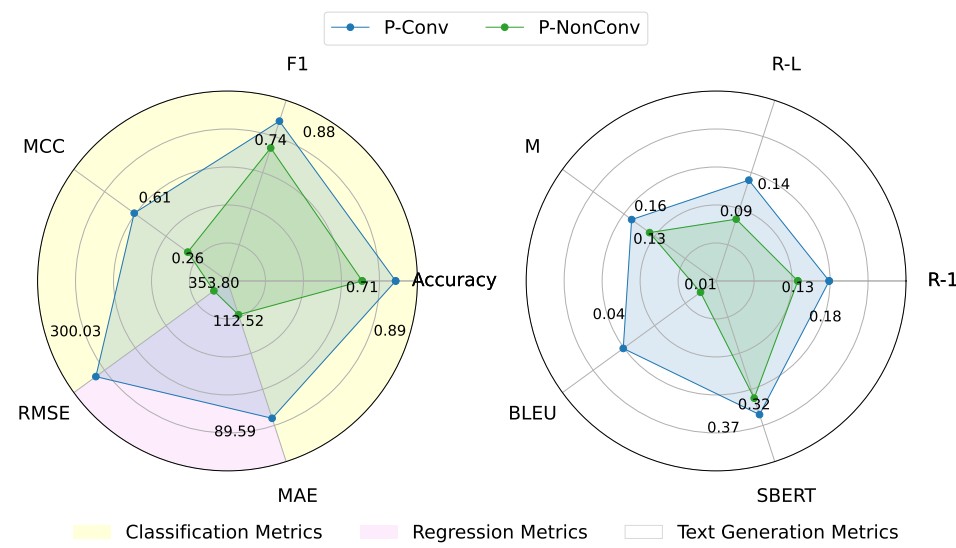

Figure 10: Performance of `DeepSeek-R1` on our personalized conversation benchmark (left radar chart for classification and regression; right one for generation).

# E   MORE EXPERIMENTAL RESULTS

## E.1   PAIRED $t$-TEST ANALYSIS

To assess the statistical significance of performance differences between personalized conversational prompting (**P-Conv**) and the baseline that omits conversational context (**P-NonConv**), we conduct paired $t$-tests across all test instances for the follow-up text generation task. For each LLM and evaluation metric, we compute the $t$-statistic over the paired prediction scores obtained under the two settings.

| | | PERFORMANCE METRICS | |
| | | (**P-Conv (Ours)** \| NP-Conv \| P-NonConv) | |
| **Task** | **Metric** | GPT-4.1 | Claude-3.5 |
|---|---|---|---|
| **Person. Conv. Senti. Classif.** (Sec. 3.2.1) | Accuracy ↑ | **0.9122** \| 0.8868 \| 0.7862 | **0.9109** \| 0.8956 \| 0.8192 |
| | F1 ↑ | **0.9481** \| 0.9336 \| 0.8720 | **0.9474** \| 0.9384 \| 0.8908 |
| | MCC ↑ | **0.6770** \| 0.5726 \| 0.2266 | **0.6721** \| 0.6113 \| 0.3666 |
| **Person. Conv. Impact Forec.** (Sec. 3.2.2) | RMSE ↓ | **310.09** \| 316.84 \| 350.29 | **282.48** \| 351.17 \| 344.75 |
| | MAE ↓ | **97.52** \| 97.83 \| 113.46 | **85.39** \| 99.25 \| 109.27 |
| **Person. Conv. Next-Text Gen.** (Sec. 3.2.3) | ROUGE-1 ↑ | **0.2777** \| 0.2137 \| 0.2248 | **0.2161** \| 0.1224 \| 0.1645 |
| | ROUGE-L ↑ | **0.2115** \| 0.1508 \| 0.1565 | **0.1719** \| 0.0812 \| 0.1130 |
| | METEOR ↑ | **0.2677** \| 0.2139 \| 0.2316 | **0.1913** \| 0.1447 \| 0.1636 |
| | BLEU ↑ | **0.0604** \| 0.0188 \| 0.0206 | **0.0509** \| 0.0063 \| 0.0123 |
| | SBERT ↑ | **0.4757** \| 0.4122 \| 0.4322 | **0.3942** \| 0.2676 \| 0.3512 |

Table 8: Results for GPT-4.1 and Claude-3.5 on the three personalized conversational benchmark tasks (More models' results in Tab. 20), computed over the entire dataset (aggregated from 10 domains). Within each cell, performance is shown as: **Personalized Conversational (P-Conv, bolded, left)** \| Non-Personalized Conversational (NP-Conv, middle) \| Personalized Non-Conversational (P-NonConv, right). "P-Conv" denotes personalized conversational performance. "NP-Conv" denotes non-personalized conversational performance, where the conversational history is randomly sampled and does not necessarily belong to the target user for contextual personalization. "P-NonConv" denotes personalized non-conversational performance. We report standard metrics for classification (Accuracy, F1, MCC), regression (RMSE, MAE), and text generation (ROUGE, METEOR, BLEU, SBERT).

| | | PERSONALIZED | | | |
| | | (Hypothesis Testing $t$-statistics for **P-Conv** vs. P-NonConv) | | | |
| **Task** | **Metric** | GPT-4.1 | GPT-4o-mini | Claude-3.5 | LLaMA3.3 | DeepSeek-R1 |
|---|---|---|---|---|---|---|
| **Person. Conv. Next-Text Gen.** (Sec. 3.2.3) | ROUGE-1 ↑ | 46.0776 | 15.1965 | 31.0044 | 39.3839 | 33.8472 |
| | ROUGE-L ↑ | 46.0989 | 10.0183 | 45.9076 | 42.2098 | 38.2525 |
| | METEOR ↑ | 15.0208 | - | 25.6402 | 14.0191 | 20.0036 |
| | BLEU ↑ | 20.3417 | 10.7500 | 21.3349 | 28.8257 | 27.5687 |
| | SBERT ↑ | 21.9320 | 17.4189 | 25.7851 | 26.1129 | 27.2352 |

Table 9: Paired $t$-test analysis comparing the personalized conversational setting (**P-Conv**) against a non-conversational baseline (**P-NonConv**) for the follow-up Text Generation task (Section 3.2.3). The table reports $t$-statistics for five evaluation metrics across five different LLMs. Only statistically significant improvements at the $\alpha = 0.01$ level are shown. Higher values indicate that P-Conv significantly outperforms P-NonConv under the corresponding setting and model.

As shown in Table 9, all reported values represent statistically significant improvements at the $\alpha = 0.01$ level. The positive $t$-values indicate that incorporating full conversational context consistently enhances model performance across all metrics and models evaluated. Notably, models such as GPT-4.1 and DeepSeek-R1 show strong gains in both lexical metrics (e.g., ROUGE and BLEU) and semantic similarity (SBERT), highlighting the effectiveness of trajectory-level personalization.

| Task | Metric | PERSONALIZED (**Conversational** \| Non-Conversational) | | | | |
|---|---|---|---|---|---|---|
| | | GPT-4.1 | GPT-4o-mini | Claude-3.5 | LLaMA3.3 | DeepSeek-R1 |
| **Person. Conv. Senti. Classif.** (Sec. 3.2.1) | Accuracy ↑ | **0.8727** \| 0.7055 | **0.6327** \| 0.5382 | **0.8869** \| 0.7527 | **0.8182** \| 0.6654 | **0.8473** \| 0.6109 |
| | F1 ↑ | **0.9206** \| 0.8103 | **0.7321** \| 0.6482 | **0.9310** \| 0.8396 | **0.8054** \| 0.6762 | **0.8426** \| 0.6402 |
| | MCC ↑ | **0.6171** \| 0.1524 | **0.1871** \| 0.0330 | **0.6604** \| 0.3001 | **0.4404** \| 0.1307 | **0.5509** \| 0.1790 |
| **Person. Conv. Impact Forec.** (Sec. 3.2.2) | RMSE ↓ | **168.72** \| 190.33 | **171.41** \| 191.41 | **153.86** \| 186.92 | **180.74** \| 190.56 | **149.45** \| 189.94 |
| | MAE ↓ | **71.96** \| 83.16 | **71.36** \| 84.62 | **61.02** \| 78.34 | **73.03** \| 81.93 | **59.83** \| 80.58 |
| **Person. Conv. Next-Text Gen.** (Sec. 3.2.3) | ROUGE-1 ↑ | **0.2788** \| 0.2255 | **0.2421** \| 0.2108 | **0.1653** \| 0.1093 | **0.2124** \| 0.1610 | **0.1801** \| 0.1381 |
| | ROUGE-L ↑ | **0.2072** \| 0.1498 | **0.1753** \| 0.1402 | **0.2140** \| 0.1661 | **0.1579** \| 0.1013 | **0.1356** \| 0.0883 |
| | METEOR ↑ | **0.2634** \| 0.2268 | **0.2033** \| 0.2139 | **0.1806** \| 0.1608 | **0.1917** \| 0.1663 | **0.1638** \| 0.1390 |
| | BLEU ↑ | **0.0658** \| 0.0212 | **0.0271** \| 0.0158 | **0.0594** \| 0.0106 | **0.0541** \| 0.0094 | **0.0422** \| 0.0072 |
| | SBERT ↑ | **0.4799** \| 0.4266 | **0.4352** \| 0.4045 | **0.3858** \| 0.3456 | **0.3756** \| 0.3363 | **0.3698** \| 0.3291 |

Table 10: Main results across the three personalized conversational benchmark tasks. Within each cell, the **Conversational performance (bolded, left)** is shown separated by a vertical line from the Non-Conversational performance (right) for the respective models. Results are computed on the **art domain**. We report standard metrics for classification (Accuracy, F1, MCC), regression (RMSE, MAE), and text generation (ROUGE, METEOR, BLEU, SBERT).

These findings reinforce the importance of leveraging contextual information beyond isolated user text snippets in order to generate more coherent and personalized responses.

## E.2 CASE STUDY

As shown in Fig. 5, we present a detailed case study of the personalized conversational follow-up text generation task. The example demonstrates how a large language model utilizes both user-specific history and full conversational context to predict the next response.

The left side illustrates the prompt construction, including a few-shot demonstration and the actual input conversation with the target reply masked. The right side shows the model's generated response, postprocessed output, and evaluation metrics compared against the ground-truth label.

This example highlights the model's ability to generate contextually coherent and user-consistent responses. While minor lexical mismatches are observed, the predicted reply preserves the intended semantics and tone of the original author. Evaluation scores confirm a reasonably good match under both lexical and semantic metrics.

## E.3 BASELINE COMPARISON

The results in Table 8 and Table. 20 reveal two key insights about the importance of personalization and conversation structure in language model performance.

First, we observe that integrating both user history and conversational trajectory (P-Conv) leads to substantial performance improvements across all three tasks—classification, regression, and generation. In particular, the performance gap between P-Conv and P-NonConv confirms that user-specific conversational dynamics cannot be captured by non-conversational signals alone. This suggests that static personalization, while helpful, is insufficient without modeling the evolving structure and context of user interaction.

Second, the NP-Conv baseline performs worse than P-NonConv in many cases, despite using conversational information. This indicates that injecting mismatched or user-agnostic conversation context can actively degrade model performance. It highlights the importance of contextual alignment: conversations are only beneficial when they are semantically and behaviorally tied to the user of interest.

| Task | Metric | PERSONALIZED (**Conversational** \| Non-Conversational) | | | | |
|------|--------|---------|------------|-----------|----------|------------|
| | | GPT-4.1 | GPT-4o-mini | Claude-3.5 | LLaMA3.3 | DeepSeek-R1 |
| **Person. Conv. Senti. Classif.** (Sec. 3.2.1) | Accuracy ↑ | **0.9187** \| 0.7895 | **0.6842** \| 0.6938 | **0.9234** \| 0.8454 | **0.8421** \| 0.7895 | **0.8900** \| 0.7560 |
| | F1 ↑ | **0.9516** \| 0.8728 | **0.7911** \| 0.7975 | **0.9545** \| 0.9059 | **0.8378** \| 0.7967 | **0.8856** \| 0.7714 |
| | MCC ↑ | **0.7130** \| 0.2641 | **0.1671** \| 0.1939 | **0.7309** \| 0.4748 | **0.4544** \| 0.3597 | **0.6149** \| 0.3203 |
| **Person. Conv. Impact Forec.** (Sec. 3.2.2) | RMSE ↓ | **521.77** \| 566.48 | **547.91** \| 567.99 | **473.42** \| 511.20 | **550.85** \| 567.41 | **503.57** \| 582.63 |
| | MAE ↓ | **96.29** \| 108.64 | **97.02** \| 110.58 | **83.24** \| 100.02 | **99.84** \| 107.74 | **86.52** \| 114.50 |
| **Person. Conv. Next-Text Gen.** (Sec. 3.2.3) | ROUGE-1 ↑ | **0.2847** \| 0.2417 | **0.2507** \| 0.2247 | **0.2204** \| 0.1777 | **0.2148** \| 0.1761 | **0.1857** \| 0.1532 |
| | ROUGE-L ↑ | **0.2045** \| 0.1581 | **0.1836** \| 0.1471 | **0.1712** \| 0.1143 | **0.1548** \| 0.1095 | **0.1344** \| 0.0969 |
| | METEOR ↑ | **0.2684** \| 0.2497 | **0.2124** \| 0.2329 | **0.1939** \| 0.1770 | **0.1851** \| 0.1837 | **0.1638** \| 0.1529 |
| | BLEU ↑ | **0.0543** \| 0.0225 | **0.0303** \| 0.0176 | **0.0500** \| 0.0121 | **0.0409** \| 0.0101 | **0.0355** \| 0.0090 |
| | SBERT ↑ | **0.5014** \| 0.4626 | **0.4559** \| 0.4333 | **0.4112** \| 0.3720 | **0.3950** \| 0.3652 | **0.3903** \| 0.3617 |

Table 11: Main results across the three personalized conversational benchmark tasks. Within each cell, the **Conversational performance (bolded, left)** is shown separated by a vertical line from the Non-Conversational performance (right) for the respective models. Results are computed on the **books domain**. We report standard metrics for classification (Accuracy, F1, MCC), regression (RMSE, MAE), and text generation (ROUGE, METEOR, BLEU, SBERT).

| Task | Metric | PERSONALIZED (**Conversational** \| Non-Conversational) | | | | |
|------|--------|---------|------------|-----------|----------|------------|
| | | GPT-4.1 | GPT-4o-mini | Claude-3.5 | LLaMA3.3 | DeepSeek-R1 |
| **Person. Conv. Senti. Classif.** (Sec. 3.2.1) | Accuracy ↑ | **0.8141** \| 0.6803 | **0.6914** \| 0.6468 | **0.8550** \| 0.6952 | **0.7993** \| 0.6952 | **0.8476** \| 0.7026 |
| | F1 ↑ | **0.8792** \| 0.7923 | **0.7701** \| 0.7397 | **0.9037** \| 0.7919 | **0.7754** \| 0.6821 | **0.8422** \| 0.7090 |
| | MCC ↑ | **0.5452** \| 0.1422 | **0.3038** \| 0.1916 | **0.6532** \| 0.2341 | **0.5004** \| 0.2387 | **0.6291** \| 0.3413 |
| **Person. Conv. Impact Forec.** (Sec. 3.2.2) | RMSE ↓ | **251.42** \| 267.50 | **251.65** \| 267.93 | **229.07** \| 266.28 | **259.81** \| 266.54 | **217.15** \| 267.38 |
| | MAE ↓ | **85.94** \| 95.06 | **86.34** \| 95.75 | **72.44** \| 91.47 | **86.91** \| 93.06 | **72.97** \| 93.42 |
| **Person. Conv. Next-Text Gen.** (Sec. 3.2.3) | ROUGE-1 ↑ | **0.2798** \| 0.2252 | **0.2552** \| 0.2127 | **0.2276** \| 0.1736 | **0.2137** \| 0.1575 | **0.1877** \| 0.1420 |
| | ROUGE-L ↑ | **0.2070** \| 0.1513 | **0.1913** \| 0.1422 | **0.1796** \| 0.1156 | **0.1607** \| 0.1001 | **0.1437** \| 0.0904 |
| | METEOR ↑ | **0.2815** \| 0.2480 | **0.2274** \| 0.2356 | **0.2030** \| 0.1755 | **0.1950** \| 0.1768 | **0.1744** \| 0.1505 |
| | BLEU ↑ | **0.0581** \| 0.0211 | **0.0311** \| 0.0176 | **0.0536** \| 0.0124 | **0.0456** \| 0.0090 | **0.0390** \| 0.0078 |
| | SBERT ↑ | **0.4745** \| 0.4335 | **0.4411** \| 0.4048 | **0.3949** \| 0.3474 | **0.3729** \| 0.3311 | **0.3667** \| 0.3236 |

Table 12: Main results across the three personalized conversational benchmark tasks. Within each cell, the **Conversational performance (bolded, left)** is shown separated by a vertical line from the Non-Conversational performance (right) for the respective models. Results are computed on the **entrepreneur domain**. We report standard metrics for classification (Accuracy, F1, MCC), regression (RMSE, MAE), and text generation (ROUGE, METEOR, BLEU, SBERT).

# F USE OF LLMS

In addition to conventional data collection and analysis, we made use of LLMs at several stages of our work. First, LLMs were applied during the writing process to assist with polishing and improving the clarity of the manuscript. Second, LLMs were also leveraged to support certain aspects of dataset construction, where they were used to generate and refine synthetic examples in a controlled manner. These uses were complementary to our primary methodology and were limited to auxiliary tasks such as language editing and expanding data diversity, without affecting the core experimental design or evaluation.

| Task | Metric | PERSONALIZED (**Conversational** \| Non-Conversational) | | | | |
|---|---|---|---|---|---|---|
| | | GPT-4.1 | GPT-4o-mini | Claude-3.5 | LLaMA3.3 | DeepSeek-R1 |
| **Person. Conv. Senti. Classif.** (Sec. 3.2.1) | Accuracy ↑ | **0.8758** \| 0.7362 | **0.7119** \| 0.7119 | **0.8694** \| 0.7933 | **0.7977** \| 0.7375 | **0.8540** \| 0.7157 |
| | F1 ↑ | **0.9180** \| 0.8322 | **0.7919** \| 0.7938 | **0.9137** \| 0.8648 | **0.7840** \| 0.7301 | **0.8485** \| 0.7165 |
| | MCC ↑ | **0.6911** \| 0.2718 | **0.3528** \| 0.3167 | **0.6532** \| 0.4577 | **0.4739** \| 0.3357 | **0.6316** \| 0.3160 |
| **Person. Conv. Impact Forec.** (Sec. 3.2.2) | RMSE ↓ | **322.29** \| 422.60 | **330.27** \| 423.93 | **303.67** \| 419.79 | **343.73** \| 423.05 | **317.46** \| 421.96 |
| | MAE ↓ | **112.66** \| 135.41 | **111.48** \| 137.41 | **100.94** \| 130.84 | **117.70** \| 135.02 | **101.05** \| 132.72 |
| **Person. Conv. Next-Text Gen.** (Sec. 3.2.3) | ROUGE-1 ↑ | **0.2782** \| 0.2283 | **0.2511** \| 0.2145 | **0.2063** \| 0.1636 | **0.1962** \| 0.1508 | **0.1719** \| 0.1338 |
| | ROUGE-L ↑ | **0.2074** \| 0.1638 | **0.1965** \| 0.1521 | **0.1674** \| 0.1175 | **0.1515** \| 0.1017 | **0.1363** \| 0.0924 |
| | METEOR ↑ | **0.2618** \| 0.2323 | **0.2114** \| 0.2228 | **0.1001** \| 0.1627 | **0.1727** \| 0.1629 | **0.1574** \| 0.1385 |
| | BLEU ↑ | **0.0498** \| 0.0218 | **0.0314** \| 0.0175 | **0.0448** \| 0.0139 | **0.0370** \| 0.0091 | **0.0354** \| 0.0089 |
| | SBERT ↑ | **0.4536** \| 0.4184 | **0.4222** \| 0.3788 | **0.3743** \| 0.3383 | **0.3557** \| 0.3149 | **0.3536** \| 0.3122 |

Table 13: Main results across the three personalized conversational benchmark tasks. Within each cell, the **Conversational performance (bolded, left)** is shown separated by a vertical line from the Non-Conversational performance (right) for the respective models. Results are computed on the **gaming domain**. We report standard metrics for classification (Accuracy, F1, MCC), regression (RMSE, MAE), and text generation (ROUGE, METEOR, BLEU, SBERT).

| Task | Metric | PERSONALIZED (**Conversational** \| Non-Conversational) | | | | |
|---|---|---|---|---|---|---|
| | | GPT-4.1 | GPT-4o-mini | Claude-3.5 | LLaMA3.3 | DeepSeek-R1 |
| **Person. Conv. Senti. Classif.** (Sec. 3.2.1) | Accuracy ↑ | **0.9535** \| 0.9030 | **0.8566** \| 0.8606 | **0.9585** \| 0.9172 | **0.9263** \| 0.8758 | **0.9383** \| 0.8636 |
| | F1 ↑ | **0.9749** \| 0.9474 | **0.9181** \| 0.9209 | **0.9776** \| 0.9549 | **0.9231** \| 0.8821 | **0.9397** \| 0.8791 |
| | MCC ↑ | **0.6793** \| 0.3362 | **0.3679** \| 0.3568 | **0.7170** \| 0.4456 | **0.5095** \| 0.3165 | **0.6373** \| 0.3809 |
| **Person. Conv. Impact Forec.** (Sec. 3.2.2) | RMSE ↓ | **327.02** \| 351.75 | **317.19** \| 352.81 | **306.81** \| 351.49 | **343.73** \| 351.85 | **319.12** \| 356.44 |
| | MAE ↓ | **98.31** \| 112.67 | **93.64** \| 115.21 | **85.36** \| 111.13 | **96.56** \| 112.73 | **87.14** \| 112.02 |
| **Person. Conv. Next-Text Gen.** (Sec. 3.2.3) | ROUGE-1 ↑ | **0.2943** \| 0.2348 | **0.2699** \| 0.2236 | **0.2315** \| 0.3708 | **0.2111** \| 0.1528 | **0.1934** \| 0.1425 |
| | ROUGE-L ↑ | **0.2334** \| 0.1718 | **0.2167** \| 0.1623 | **0.1906** \| 0.1221 | **0.1678** \| 0.1043 | **0.1564** \| 0.1004 |
| | METEOR ↑ | **0.2841** \| 0.2471 | **0.2308** \| 0.2202 | **0.2036** \| 0.1726 | **0.1865** \| 0.1654 | **0.1793** \| 0.1484 |
| | BLEU ↑ | **0.0622** \| 0.0242 | **0.0426** \| 0.0201 | **0.0553** \| 0.0138 | **0.0488** \| 0.0097 | **0.0453** \| 0.0103 |
| | SBERT ↑ | **0.4917** \| 0.4476 | **0.4593** \| 0.4097 | **0.4221** \| 0.3708 | **0.3900** \| 0.3443 | **0.3941** \| 0.3512 |

Table 14: Main results across the three personalized conversational benchmark tasks. Within each cell, the **Conversational performance (bolded, left)** is shown separated by a vertical line from the Non-Conversational performance (right) for the respective models. Results are computed on the **life domain**. We report standard metrics for classification (Accuracy, F1, MCC), regression (RMSE, MAE), and text generation (ROUGE, METEOR, BLEU, SBERT).

| | | PERSONALIZED | | | | |
| | | (**Conversational** \| Non-Conversational) | | | | |
| **Task** | **Metric** | GPT-4.1 | GPT-4o-mini | Claude-3.5 | LLaMA3.3 | DeepSeek-R1 |
|---|---|---|---|---|---|---|
| **Person. Conv.** **Senti. Classif.** (Sec. 3.2.1) | Accuracy ↑ | **0.9275** \| 0.8101 | **0.7353** \| 0.7192 | **0.9190** \| 0.9132 | **0.8412** \| 0.7572 | **0.9079** \| 0.7618 |
| | F1 ↑ | **0.9568** \| 0.8875 | **0.8260** \| 0.8135 | **0.9521** \| 0.8935 | **0.8365** \| 0.7698 | **0.9062** \| 0.7788 |
| | MCC ↑ | **0.7426** \| 0.2887 | **0.2998** \| 0.2789 | **0.7103** \| 0.4782 | **0.4405** \| 0.2856 | **0.6823** \| 0.3467 |
| **Person. Conv.** **Impact Forec.** (Sec. 3.2.2) | RMSE ↓ | **251.09** \| 287.45 | **244.86** \| 288.76 | **202.03** \| 282.99 | **278.68** \| 287.35 | **215.91** \| 301.30 |
| | MAE ↓ | **94.31** \| 108.45 | **90.92** \| 110.64 | **72.22** \| 102.99 | **100.70** \| 108.09 | **78.22** \| 109.85 |
| **Person. Conv.** **Next-Text Gen.** (Sec. 3.2.3) | ROUGE-1 ↑ | **0.2684** \| 0.2247 | **0.2450** \| 0.2104 | **0.2037** \| 0.1648 | **0.1951** \| 0.1545 | **0.1661** \| 0.1362 |
| | ROUGE-L ↑ | **0.2015** \| 0.1547 | **0.1863** \| 0.1445 | **0.1580** \| 0.1125 | **0.1457** \| 0.1004 | **0.1268** \| 0.0905 |
| | METEOR ↑ | **0.2607** \| 0.2367 | **0.2105** \| 0.2252 | **0.1014** \| 0.1668 | **0.1753** \| 0.1726 | **0.1538** \| 0.1452 |
| | BLEU ↑ | **0.0522** \| 0.0205 | **0.0317** \| 0.0168 | **0.0451** \| 0.0125 | **0.0374** \| 0.0092 | **0.0335** \| 0.0082 |
| | SBERT ↑ | **0.4691** \| 0.4354 | **0.4346** \| 0.4012 | **0.3808** \| 0.3544 | **0.3642** \| 0.3383 | **0.3544** \| 0.3300 |

Table 15: Main results across the three personalized conversational benchmark tasks. Within each cell, the **Conversational performance (bolded, left)** is shown separated by a vertical line from the Non-Conversational performance (right) for the respective models. Results are computed on the **movies domain**. We report standard metrics for classification (Accuracy, F1, MCC), regression (RMSE, MAE), and text generation (ROUGE, METEOR, BLEU, SBERT).

| | | PERSONALIZED | | | | |
| | | (**Conversational** \| Non-Conversational) | | | | |
| **Task** | **Metric** | GPT-4.1 | GPT-4o-mini | Claude-3.5 | LLaMA3.3 | DeepSeek-R1 |
|---|---|---|---|---|---|---|
| **Person. Conv.** **Senti. Classif.** (Sec. 3.2.1) | Accuracy ↑ | **0.9382** \| 0.7935 | **0.5463** \| 0.4607 | **0.9298** \| 0.8427 | **0.8553** \| 0.6868 | **0.9003** \| 0.6278 |
| | F1 ↑ | **0.9654** \| 0.8784 | **0.6760** \| 0.5862 | **0.9605** \| 0.9094 | **0.8615** \| 0.7370 | **0.9049** \| 0.6931 |
| | MCC ↑ | **0.6825** \| 0.2072 | **0.0969** \| 0.0608 | **0.6467** \| 0.3165 | **0.3751** \| 0.1313 | **0.5761** \| 0.1797 |
| **Person. Conv.** **Impact Forec.** (Sec. 3.2.2) | RMSE ↓ | **367.21** \| 407.40 | **371.91** \| 408.70 | **326.02** \| 404.32 | **365.87** \| 407.55 | **341.57** \| 406.78 |
| | MAE ↓ | **120.92** \| 134.97 | **121.48** \| 138.43 | **105.36** \| 130.32 | **122.09** \| 135.53 | **118.61** \| 133.20 |
| **Person. Conv.** **Next-Text Gen.** (Sec. 3.2.3) | ROUGE-1 ↑ | **0.2670** \| 0.2082 | **0.2309** \| 0.1966 | **0.2104** \| 0.1495 | **0.2037** \| 0.1462 | **0.1692** \| 0.1218 |
| | ROUGE-L ↑ | **0.2034** \| 0.1426 | **0.1734** \| 0.1354 | **0.1680** \| 0.1003 | **0.1570** \| 0.0940 | **0.1342** \| 0.0805 |
| | METEOR ↑ | **0.2526** \| 0.2046 | **0.1905** \| 0.1955 | **0.1901** \| 0.1454 | **0.1844** \| 0.1536 | **0.1568** \| 0.1220 |
| | BLEU ↑ | **0.0705** \| 0.0163 | **0.0299** \| 0.0139 | **0.0703** \| 0.0109 | **0.0640** \| 0.0075 | **0.0520** \| 0.0067 |
| | SBERT ↑ | **0.4643** \| 0.4154 | **0.4193** \| 0.3809 | **0.3823** \| 0.3324 | **0.3661** \| 0.3240 | **0.3609** \| 0.3152 |

Table 16: Main results across the three personalized conversational benchmark tasks. Within each cell, the **Conversational performance (bolded, left)** is shown separated by a vertical line from the Non-Conversational performance (right) for the respective models. Results are computed on the **politics domain**. We report standard metrics for classification (Accuracy, F1, MCC), regression (RMSE, MAE), and text generation (ROUGE, METEOR, BLEU, SBERT).

| Task | Metric | PERSONALIZED (**Conversational** \| Non-Conversational) | | | | |
| | | GPT-4.1 | GPT-4o-mini | Claude-3.5 | LLaMA3.3 | DeepSeek-R1 |
| --- | --- | --- | --- | --- | --- | --- |
| **Person. Conv. Senti. Classif.** (Sec. 3.2.1) | Accuracy ↑ | **0.8957** \| 0.7174 | **0.6043** \| 0.5913 | **0.8783** \| 0.7174 | **0.8304** \| 0.6870 | **0.8174** \| 0.6376 |
| | F1 ↑ | **0.9351** \| 0.8159 | **0.7074** \| 0.6948 | **0.9239** \| 0.8127 | **0.8304** \| 0.7038 | **0.8310** \| 0.6681 |
| | MCC ↑ | **0.6759** \| 0.2102 | **0.1592** \| 0.1448 | **0.6240** \| 0.2436 | **0.5053** \| 0.2012 | **0.5279** \| 0.2314 |
| **Person. Conv. Impact Forec.** (Sec. 3.2.2) | RMSE ↓ | **276.02** \| 289.46 | **271.00** \| 290.53 | **259.62** \| 287.14 | **284.18** \| 289.17 | **265.02** \| 288.90 |
| | MAE ↓ | **80.67** \| 89.55 | **78.87** \| 91.52 | **68.97** \| 84.87 | **80.60** \| 88.84 | **71.08** \| 87.69 |
| **Person. Conv. Next-Text Gen.** (Sec. 3.2.3) | ROUGE-1 ↑ | **0.2872** \| 0.2282 | **0.2502** \| 0.2143 | **0.2254** \| 0.1763 | **0.2233** \| 0.1622 | **0.1901** \| 0.1460 |
| | ROUGE-L ↑ | **0.2173** \| 0.1536 | **0.1870** \| 0.1439 | **0.1783** \| 0.1161 | **0.1704** \| 0.1036 | **0.1459** \| 0.0951 |
| | METEOR ↑ | **0.2888** \| 0.2370 | **0.2171** \| 0.2246 | **0.2056** \| 0.1761 | **0.2048** \| 0.1749 | **0.1794** \| 0.1499 |
| | BLEU ↑ | **0.0710** \| 0.0197 | **0.0341** \| 0.0168 | **0.0670** \| 0.0126 | **0.0633** \| 0.0091 | **0.0520** \| 0.0081 |
| | SBERT ↑ | **0.4990** \| 0.4484 | **0.4615** \| 0.4165 | **0.4138** \| 0.3691 | **0.3954** \| 0.3492 | **0.3855** \| 0.3458 |

Table 17: Main results across the three personalized conversational benchmark tasks. Within each cell, the **Conversational performance (bolded, left)** is shown separated by a vertical line from the Non-Conversational performance (right) for the respective models. Results are computed on the **science domain**. We report standard metrics for classification (Accuracy, F1, MCC), regression (RMSE, MAE), and text generation (ROUGE, METEOR, BLEU, SBERT).

| Task | Metric | PERSONALIZED (**Conversational** \| Non-Conversational) | | | | |
| | | GPT-4.1 | GPT-4o-mini | Claude-3.5 | LLaMA3.3 | DeepSeek-R1 |
| --- | --- | --- | --- | --- | --- | --- |
| **Person. Conv. Senti. Classif.** (Sec. 3.2.1) | Accuracy ↑ | **0.8944** \| 0.7325 | **0.5809** \| 0.5315 | **0.8876** \| 0.7338 | **0.8041** \| 0.6627 | **0.8535** \| 0.6098 |
| | F1 ↑ | **0.9375** \| 0.8353 | **0.7036** \| 0.6497 | **0.9335** \| 0.8340 | **0.7990** \| 0.6932 | **0.8565** \| 0.6524 |
| | MCC ↑ | **0.6105** \| 0.1253 | **0.0543** \| 0.0523 | **0.5841** \| 0.1641 | **0.3057** \| 0.1219 | **0.5270** \| 0.0944 |
| **Person. Conv. Impact Forec.** (Sec. 3.2.2) | RMSE ↓ | **280.00** \| 300.47 | **271.06** \| 301.76 | **257.96** \| 297.42 | **290.93** \| 300.35 | **263.41** \| 299.60 |
| | MAE ↓ | **95.27** \| 105.68 | **93.10** \| 108.30 | **83.43** \| 101.10 | **95.33** \| 105.49 | **86.16** \| 103.77 |
| **Person. Conv. Next-Text Gen.** (Sec. 3.2.3) | ROUGE-1 ↑ | **0.2796** \| 0.2160 | **0.2394** \| 0.2050 | **0.2149** \| 0.1569 | **0.2059** \| 0.1524 | **0.1801** \| 0.1291 |
| | ROUGE-L ↑ | **0.2132** \| 0.1479 | **0.1789** \| 0.1399 | **0.1704** \| 0.1060 | **0.1574** \| 0.0991 | **0.1409** \| 0.0855 |
| | METEOR ↑ | **0.2661** \| 0.2168 | **0.2023** \| 0.2073 | **0.1912** \| 0.1528 | **0.1853** \| 0.1602 | **0.1674** \| 0.1299 |
| | BLEU ↑ | **0.0708** \| 0.0180 | **0.0305** \| 0.0154 | **0.0631** \| 0.0107 | **0.0551** \| 0.0082 | **0.0507** \| 0.0071 |
| | SBERT ↑ | **0.4791** \| 0.4223 | **0.4319** \| 0.3905 | **0.3950** \| 0.3361 | **0.3693** \| 0.3260 | **0.3704** \| 0.3244 |

Table 18: Main results across the three personalized conversational benchmark tasks. Within each cell, the **Conversational performance (bolded, left)** is shown separated by a vertical line from the Non-Conversational performance (right) for the respective models. Results are computed on the **technology domain**. We report standard metrics for classification (Accuracy, F1, MCC), regression (RMSE, MAE), and text generation (ROUGE, METEOR, BLEU, SBERT).

| | | PERSONALIZED (**Conversational** \| Non-Conversational) | | | | |
|---|---|---|---|---|---|---|
| **Task** | **Metric** | GPT-4.1 | GPT-4o-mini | Claude-3.5 | LLaMA3.3 | DeepSeek-R1 |
| **Person. Conv. Senti. Classif.** (Sec. 3.2.1) | Accuracy ↑ | **0.9294** \| 0.7699 | **0.5920** \| 0.5153 | **0.9264** \| 0.8037 | **0.8589** \| 0.6411 | **0.8712** \| 0.5490 |
| | F1 ↑ | **0.9603** \| 0.8619 | **0.7127** \| 0.6393 | **0.9585** \| 0.8806 | **0.8666** \| 0.7018 | **0.8800** \| 0.6246 |
| | MCC ↑ | **0.6494** \| 0.1953 | **0.1824** \| 0.1246 | **0.6371** \| 0.3687 | **0.4244** \| 0.1384 | **0.4989** \| 0.1214 |
| **Person. Conv. Impact Forec.** (Sec. 3.2.2) | RMSE ↓ | 257.22 \| 268.26 | 258.80 \| 269.83 | 249.99 \| 269.30 | 260.40 \| 268.68 | 258.96 \| 267.99 |
| | MAE ↓ | **96.35** \| 104.35 | **95.73** \| 106.97 | **87.44** \| 101.29 | **93.78** \| 104.52 | **96.48** \| 102.32 |
| **Person. Conv. Next-Text Gen.** (Sec. 3.2.3) | ROUGE-1 ↑ | **0.2712** \| 0.2148 | **0.2375** \| 0.2049 | **0.2087** \| 0.1550 | **0.2067** \| 0.1492 | **0.1707** \| 0.1272 |
| | ROUGE-L ↑ | **0.2045** \| 0.1485 | **0.1776** \| 0.1408 | **0.1634** \| 0.1062 | **0.1584** \| 0.0965 | **0.1333** \| 0.0845 |
| | METEOR ↑ | **0.2571** \| 0.2089 | **0.1992** \| 0.2034 | **0.1847** \| 0.1491 | **0.1875** \| 0.1554 | **0.1594** \| 0.1250 |
| | BLEU ↑ | **0.0628** \| 0.0173 | **0.0284** \| 0.0145 | **0.0561** \| 0.0102 | **0.0526** \| 0.0078 | **0.0423** \| 0.0068 |
| | SBERT ↑ | **0.4780** \| 0.4271 | **0.4363** \| 0.3952 | **0.3942** \| 0.3551 | **0.3780** \| 0.3368 | **0.3699** \| 0.3311 |

Table 19: Main results across the three personalized conversational benchmark tasks. Within each cell, the **Conversational performance (bolded, left)** is shown separated by a vertical line from the Non-Conversational performance (right) for the respective models. Results are computed on the **worldnews domain**. We report standard metrics for classification (Accuracy, F1, MCC), regression (RMSE, MAE), and text generation (ROUGE, METEOR, BLEU, SBERT).

| | | PERFORMANCE METRICS (**P-Conv (Ours)** \| NP-Conv \| P-NonConv) | |
|---|---|---|---|
| **Task** | **Metric** | LLaMA3.3 | DeepSeek-R1 |
| **Person. Conv. Senti. Classif.** (Sec. 3.2.1) | Accuracy ↑ | **0.8458** \| 0.8215 \| 0.7305 | **0.8853** \| 0.8584 \| 0.7092 |
| | F1 ↑ | **0.8401** \| 0.8099 \| 0.7495 | **0.8848** \| 0.8548 \| 0.7362 |
| | MCC ↑ | **0.4420** \| 0.3273 \| 0.2333 | **0.6070** \| 0.4967 \| 0.2586 |
| **Person. Conv. Impact Forec.** (Sec. 3.2.2) | RMSE ↓ | **319.83** \| 330.83 \| 350.43 | **300.03** \| 551.67 \| 353.80 |
| | MAE ↓ | **101.25** \| 109.05 \| 113.18 | **89.59** \| 111.09 \| 112.52 |
| **Person. Conv. Next-Text Gen.** (Sec. 3.2.3) | ROUGE-1 ↑ | **0.2055** \| 0.1431 \| 0.1540 | **0.1786** \| 0.1152 \| 0.1359 |
| | ROUGE-L ↑ | **0.1572** \| 0.0955 \| 0.1009 | **0.1395** \| 0.0815 \| 0.0911 |
| | METEOR ↑ | **0.1838** \| 0.1520 \| 0.1659 | **0.1649** \| 0.1135 \| 0.1401 |
| | BLEU ↑ | **0.0480** \| 0.0082 \| 0.0089 | **0.0423** \| 0.0077 \| 0.0083 |
| | SBERT ↑ | **0.3733** \| 0.3055 \| 0.3339 | **0.3699** \| 0.2754 \| 0.3307 |

Table 20: Results for LLaMA3.3 and DeepSeek-R1 on the three personalized conversational benchmark tasks, computed over the entire dataset (aggregated from 10 domains). Within each cell, performance is shown as: **Personalized Conversational (P-Conv, bolded, left)** | Non-Personalized Conversational (NP-Conv, middle) | Personalized Non-Conversational (P-NonConv, right). "P-Conv" denotes personalized conversational performance. "NP-Conv" denotes non-personalized conversational performance, where the conversational history is randomly sampled and does not necessarily belong to the target user for contextual personalization. (Note: NP-Conv data is marked as '-' for these models in the provided source). "P-NonConv" denotes personalized non-conversational performance. We report standard metrics for classification (Accuracy, F1, MCC), regression (RMSE, MAE), and text generation (ROUGE, METEOR, BLEU, SBERT). The section references (e.g., Sec. 3.2.1) are kept from the original for context but would require corresponding labels in your document to link correctly.