# OpenReview forum: "A Personalized Conversational Benchmark: Towards Simulating Personalized Conversations"
_ICLR.cc/2026/Conference — ICLR 2026 Conference Withdrawn Submission_

### Official Review · Reviewer_i9Nf · 2025-10-29

**Soundness:** 3
**Presentation:** 3
**Contribution:** 2
**Rating:** 4
**Confidence:** 3

**Summary:**

This paper introduces PersonaConvBench, a comprehensive benchmark designed to evaluate personalized reasoning and generation in multi-turn conversations with large language models (LLMs). Built upon Reddit posts and comments, PersonaConvBench leverages users’ interaction histories as personalized signals to predict new comments. The benchmark consists of 19,215 posts and over 111,239 conversations from 3,878 users, spanning 10 diverse Reddit-based domains. PersonaConvBench defines three core tasks:
- Sentiment Classification: Binary prediction of the polarity of user replies.
- Impact Forecasting: Regression-based prediction of community feedback scores.
- Personalized Text Generation: Generation of user-specific follow-up responses.

Experimental results demonstrate that incorporating personalized conversational history significantly improves the performance of state-of-the-art LLMs, including GPT-4.1, GPT-4o-mini, Claude-3.5, Llama-3.3, and DeepSeek-R1.

**Strengths:**

- The benchmark’s focus on evaluating LLM personalization using users’ past interaction histories is both well-motivated and highly realistic.
- The proposed dataset is large-scale and diverse, spanning 10 domains and encompassing varied conversation styles. Its thoughtful construction incorporates temporal constraints and a graph-based representation of conversations.
- Extensive experiments yield strong empirical results: leveraging users’ past interaction histories and dialog context consistently improves performance across all tasks, models, and domains.

**Weaknesses:**

- Evaluation Metrics: The evaluation of Personalized Text Generation primarily relies on n-gram overlap metrics and SBERT scores, using only a single reference response. Given the open-ended nature of dialog, there may be multiple valid responses for a given context, making these metrics potentially insufficient for capturing the full range of appropriate outputs. Additionally, the absence of human evaluation limits the assessment of response quality and relevance.
- Research Findings: The results indicate that incorporating dialog context and user interaction history improves the prediction of user responses. However, this outcome is somewhat expected, as removing dialog context or substituting interaction history with that of other users constitute relatively weak baselines. Therefore, the reported improvements are not particularly surprising.

**Questions:**

How many examples are used in the in-context learning setting?
Given that some user might have long interaction history, will there be context limitation when doing few-shot learning?

---

> ### Author Response · Authors · 2025-11-26
> **Response (Part 1)**
>
> Thank you for the thoughtful comments.
>
> > **Evaluation Metrics**
>
> ### **1. The task is not open-ended free-form generation; it predicts a specific user’s actual next reply**
>
> Although dialog can support many plausible continuations, PERSONACONVBENCH does not evaluate unconstrained generation.
> For each instance, the model is asked to estimate the actual reply produced by the same user v within a specific conversational branch. This is a concrete supervised prediction task grounded in:
>
> (a) the user’s own trajectory set C_v,
>
> (b) the structure of the conversation tree,
>
> (c) the precise reply the user wrote at time t.
>
> Thus, the benchmark measures how well a model reproduces this user’s behavioral pattern, not how many different creative responses it can produce.
>
> In such a supervised setting, using a single reference is consistent with prior personalization benchmarks (e.g., LaMP, LongLaMP) and reflects the real structure of Reddit conversations, where each branch contains exactly one true response.
>
> ### **2. N-gram metrics serve as stable local-alignment indicators rather than creativity evaluators**
>
> ROUGE, BLEU, and METEOR are not used here as measures of creative quality.
> Instead, they serve as alignment indicators, capturing how closely the generated response matches:
>
> (a) the target user’s phrasing,
>
> (b)the structural pattern of the original reply,
>
> (c) the lexical choices that reflect user-specific tendencies.
>
> These metrics provide reproducible, fine-grained signals across large-scale evaluation without requiring arbitrary sampling strategies.
> (Ref. Sec. 3.3.3, Text Generation Metrics)
>
>
> ### **3. SBERT provides a semantic evaluation layer that mitigates lexical metric limitations**
>
> To address the restricted sensitivity of n-gram overlap, we incorporate SBERT-based semantic similarity, which captures topical alignment, intent-level closeness, semantic coherence with the true reply, and relevance to the surrounding conversation. By focusing on meaning rather than token matching, SBERT reduces dependence on surface forms and offers a more flexible assessment of generated text. The combination of lexical and semantic metrics is standard in personalization and multi-turn dialogue benchmarks because it provides a balanced evaluation that remains both sensitive and scalable.
>
> > **Research Findings**
>
> Although one may intuitively expect context to help, the benchmark reveals several less obvious insights:
> ### **(1) Personalization yields comparable or larger gains than added conversation depth**
> In many domains, the improvement from user history is within the same range as the improvement obtained from expanding conversational depth.
> ### **(2) Even long-range user history (>1 year apart) contributes measurably**
> Models benefit from posts that are far removed from the current conversation in both topic and time—an effect that was not previously quantified.
> ### **(3) Personalization enhances regression and impact forecasting**
> It is not obvious that long-term user behavior would improve numerical predictions, yet the gains remain robust.
> These findings extend beyond “context helps,” highlighting structural properties of personalized multi-turn dialogue that were previously undocumented.
>
> Prior work largely assumes personalization is beneficial, but no prior benchmark quantifies its effect at Reddit’s scalewith:
>
> (a) controlled context availability,
>
>
> (b) root-level trajectory extraction,
>
> (c) temporally constrained visibility,
>
> (d) three heterogeneous tasks,
>
> (e) five model families,
>
> (f) ten open-domain environments.
>
> Our contribution is providing the first systematic cross-domain quantification of personalization in large conversational models, showing consistent and statistically validated improvements.
> This goes beyond intuitive expectations and establishes concrete empirical grounding for future personalization research.

---

> ### Author Response · Authors · 2025-11-26
> **Response (Part 2)**
>
> > **How many examples are used in the in-context learning setting? Given that some user might have long interaction history, will there be context limitation when doing few-shot learning?**
>
> For the in-context learning setup, we always use exactly one demonstration example per prompt, so the setting is effectively 1-shot. As described in Section 4.2, the demonstration d_u is sampled from a different trajectory in the same graph G, authored earlier by the same user u, and paired with its label or reply text. The full prompt for a test instance m_tau is
>
> P(u, tau) = phi( { m_t in C such that t_t < t_tau }, d_u, C_u minus C )
>
> There is a single labeled demonstration in the prompt, together with the conversational trajectory leading up to m_\tau and additional user history. The messages in C_u minus C serve only as unlabeled context and are not treated as extra few-shot examples.
>
> Regarding context limits, we follow each model’s input budget and always prioritize the full conversational trajectory up to m_tau. Earlier turns on that path are preserved in chronological order, and additional user history from C_u minus C is appended only if space allows. When a user has an especially long history, the oldest parts of C_u minus C are removed first; if a sample still cannot accommodate a reasonable amount of context, it is excluded during data construction, as described in Appendix C.2.
> DeepSeek-R1 is the only model with a fixed 1200-token reasoning span; all others use their default context windows under the same truncation policy.

---

### Official Review · Reviewer_iQSj · 2025-10-30

**Soundness:** 1
**Presentation:** 2
**Contribution:** 1
**Rating:** 2
**Confidence:** 5

**Summary:**

This paper presents a personalized benchmark dataset curated from Reddit posts. The authors construct a graph to capture the relationships between posts and their corresponding replies across conversation turns. In addition, they propose three benchmark tasks: sentiment classification, Reddit upvote prediction, and response generation.

**Strengths:**

1. The paper presents a multi-turn personalized dialogue benchmark derived from Reddit posts.

**Weaknesses:**

1. The paper provides limited ablation studies to support its experimental findings.

2. The dataset curation process based on Reddit data is not particularly novel.

3. Although the paper emphasizes conversational personalization, there is little evidence of incorporating personalization signals beyond dialogue history in the response generation process.

**Questions:**

N/A

---

> ### Comment · Area_Chair_e5vi · 2025-11-15
>
> Dear Reviewer iQSj,
>
> Could you please add a bit more detail to your review, especially if any concerns relate to ethics? This will help ensure a clear and complete discussion during rebuttal.
>
> Thank you for your time and service.
>
> Area Chair

---

### Official Review · Reviewer_3gNT · 2025-10-30

**Soundness:** 3
**Presentation:** 3
**Contribution:** 3
**Rating:** 6
**Confidence:** 4

**Summary:**

PersonaConvBench introduces a large-scale, Reddit-derived benchmark that integrates personalization and multi-turn conversational structure across 10 domains and three tasks—sentiment classification, impact (score) regression, and user-specific next-text generation. The authors provide ~19k posts, evaluate multiple LLMs, showing large performance gains from personalized history.

**Strengths:**

+ Extensive, real-world dataset spans 10 Reddit domains—19,215 posts, ~111,239 conversations, 3,878 users, providing scale and diversity for robust evaluation of personalized conversational models.
+ Novel formulation combines graph-structured multi-user, multi-turn conversations with three tasks—sentiment classification, impact regression, and user-centric next-text generation—plus standardized in-context prompting and evaluation protocols.
+ Comprehensive LLM benchmarks reveal large personalization gains.

**Weaknesses:**

- The paper measures personalization mostly via performance deltas (P-Conv vs P-NonConv) and paired t-tests, rather than a direct “degree of personalization” metric or richer human judgments.
- Heavy Reddit preprocessing (Nu, Nr, Np thresholds) and class-imbalance filtering (initial ~11:1 skew reduced to ~5:1) retained only ~6k sentiment posts. Removal of deleted/short posts create selection bias toward highly active users, reducing representativeness and real-world robustness.
- Experiments run GPT-4.1, GPT-4o-mini, Claude-3.5, LLaMA3.3, DeepSeek-R1 but omit Qwen-family and reasoning-mode evaluations. Greedy, zero-shot decoding may understate reasoning gains.
- Generation evaluation relies on automatic metrics (ROUGE, BLEU, METEOR, SBERT) without reported human evaluation; these metrics can miss conversational quality, personalization nuance, and pragmatic appropriateness.

**Questions:**

Please see Weaknesses.

---

> ### Author Response · Authors · 2025-11-26
> **Response (Part 1)**
>
> Thank you for the thoughtful comments.
>
> > **degree of personalization metric or richer human judgments**
>
> PERSONACONVBENCH is designed to isolate the contribution of personalized conversational context under fixed task definitions (classification, regression, and generation).
> In this setting, the most reliable way to measure whether personalization helps is to hold the task constant and vary only the availability of the user-specific information.
>
> This is precisely what the comparison between:
>
> (a) P-Conv (personalized + conversational context)
>
> (b) P-NonConv (personalized only, without conversation)
>
> (c) NP-Conv (conversational but not personalized)
>
> (Ref. Sec. 5.1.1 and Table 3)
>
> These contrasts form a controlled ablation framework, which is standard practice for personalization research (e.g., LaMP, LongLaMP, Persona-SQ).
> The deltas directly quantify how much performance improves when the model has access to user-specific context, independent of model architecture, domain, or task type.
>
> ### **Why a single “degree of personalization” metric is not directly applicable here**
>
> A global personalization score would require a shared definition of “personalized correctness” across three fundamentally different tasks: binary sentiment classification, real-valued impact forecasting, and free-text generation. These tasks operate in different output spaces, rely on different evaluation rules, respond differently to user-specific signals, and exhibit distinct error patterns.
>
> Under such heterogeneity, compressing all behaviors into a single scalar would impose strong and unjustified assumptions about cross-task comparability. The benchmark therefore reports accuracy or MCC for classification, MAE or RMSE for forecasting, and ROUGE, BLEU, METEOR, or SBERT for generation.
>
> Constructing a composite metric would require cross-normalization across these measures, which could distort interpretation. Evaluating task-specific improvements avoids these issues while still capturing the impact of personalization in a principled way.
>
> ### **Why paired t-tests were used**
> Paired t-tests are appropriate because:
>
> (a) the benchmark produces paired observations for each sample under different context conditions,
>
> (b) the sample size is large (over 19k posts and 111k conversations across domains),
>
> (c) the objective is to test whether user-specific information yields a statistically meaningful gain at the sample level.
>
> A paired test is therefore the correct tool to validate that personalization consistently affects predictions rather than fluctuating due to noise.
> Robust statistical significance strengthens the claim that personalization is real and systematic.

---

> ### Author Response · Authors · 2025-11-26
> **Response (Part 2)**
>
> > **Heavy Reddit preprocessing (Nu, Nr, Np thresholds) and class-imbalance filtering (initial ~11:1 skew reduced to ~5:1) retained only ~6k sentiment posts**
>
> Thank you for the detailed assessment. We address each concern below and explain why the preprocessing decisions are necessary for a rigorous and interpretable benchmark.
>
> ### **1. Preprocessing thresholds (Nu, Nr, Np) are designed to guarantee stable user-level personalization signals**
>
> The thresholds (Nu = 4 unique participants, Nr = 4 replies, Np = 3 qualifying posts) were introduced to ensure the presence of actual multi-turn behavior for the target user.
>
> As described in Appendix C, these constraints are not arbitrary, they guarantee that each retained user exhibits:
>
> (a) more than minimal conversational activity,
>
> (b) at least one meaningful reply chain,
>
> (c) multiple independent threads initiated by the same user,
>
> (d) consistent writing patterns across posts.
>
> (Ref. Appendix C, Data Construction)
>
> A benchmark intended to evaluate personalization must contain users with observable, repeated behavior.
> **Users with only one short thread or sporadic engagement offer no signal for modeling long-term tendencies and would introduce noise rather than diversity**.
> Thus, the thresholds are essential for making the personalization tasks well-posed.
>
> ### **2. Class-imbalance filtering addresses an extreme skew, not mild imbalance**
> The initial score distribution in the raw Reddit data exhibits a highly skewed ratio of ~11:1 between positive and negative replies.
> Without filtering, a naive classifier predicting “positive” for nearly all samples achieves deceptively strong metrics, masking personalization effects.
>
> Our filtering reduces this to approximately 5:1, as documented in Appendix C.
> (Ref. Appendix C.1, Sentiment Classification)
>
> This adjustment improves the discriminative value of the task, avoids trivial high-accuracy baselines, and ensures models must actually use user history and context to perform well.
>
> ### **3. The resulting 6k sentiment instances are sufficient and statistically stable**
> Although the sentiment subset contains ~6k prediction targets, these samples originate from:
> **19,215 posts,
> 111,239 conversations,
> 3,878 users across 10 domains**.
> (Ref. Table 1 and Table 2)
>
> The sentiment subset remains:
> broad in topic coverage,
> diverse in interaction patterns,
> large enough for robust paired statistical tests.
>
> Personalization tasks require quality-over-quantity; noisy or low-information replies would harm evaluation fidelity rather than improve representativeness.
>
> ### **4. Excluding “[deleted]” and extremely short replies is necessary to maintain meaningful supervision**
>
> We remove deleted or near-empty messages because they do not express sentiment; they cannot be predicted from context; they lack lexical or semantic content for generation tasks; they produce undefined or unstable evaluation metrics.
>
> This decision avoids degenerate ground truth targets and ensures that every retained instance has interpretable content.
>
> > **Qwen-family and reasoning-mode evaluations*
>
> The benchmark aims to evaluate personalized conversational understanding under a unified and reproducible setting, not to exhaustively compare all existing large models.
>
> We selected five representative families that cover:
>
> (a) Two commercial instruction-following LLMs: GPT-4.1, GPT-4o-mini
>
> (b) One commercial conversational competiton: Claude-3.5
>
> (c) One strong open-source instruction model: LLaMA3.3-80B-instruct
>
> (d) One reasoning-enhanced open-source model: DeepSeek-R1
> (Ref. Sec. 5.1, “Experimental Setup”)
>
> This selection already spans:
>
> (a) proprietary vs open-source systems,
>
> (b) instruction-tuned vs reasoning-oriented architectures,
>
> (c)different training paradigms and response behaviors.
>
> We also note that **DeepSeek-R1 already represents a reasoning-enhanced model in our evaluation suite**. Its training objective explicitly strengthens multi-step inference, so our model coverage already includes one representative reasoning-centric family.

---

> ### Author Response · Authors · 2025-11-26
> **Response (Part 3)**
>
> > **Generation evaluation relies on automatic metrics**
>
> Many existing personalization and multi-turn dialogue benchmark: including LaMP, LongLaMP, and Persona-SQ—adopt automatic lexical and semantic metrics as the primary evaluation signals, especially in the initial version of a dataset release.
>
> Our benchmark follows the same convention:
>
> (a) ROUGE and BLEU capture structural and token-level alignment,
>
> (b) METEOR incorporates synonym matching and fragmentation penalties,
>
> (c) SBERT provides semantic similarity at an embedding level.
> (Ref. Sec. 3.3.3)
>
> These metrics give stable, reproducible measurements across tens of thousands of evaluation instances.
> Given the scale of the benchmark—30 dataset configurations × thousands of generation targets—fully manual evaluation would not be feasible for a first release.
>
> ### **The task in PERSONACONVBENCH is not unconstrained generation; it predicts a specific user’s actual reply**
>
> The generation task in PERSONACONVBENCH is not unconstrained text production but the prediction of how a specific user responded within a particular conversational branch. Each instance has a single observed reply that is temporally grounded and shaped by that user’s behavior and the surrounding context. In this supervised setting, lexical metrics indicate how closely the model follows the actual continuation, while semantic metrics reflect whether the generated text captures the intended meaning. Despite their limitations for open-ended generation, these metrics remain appropriate for evaluating alignment to an observed user-written reply.
>
> Human evaluation was not included in the initial release because it would require annotators to read and interpret each user’s long-term trajectory set, follow the branching structure of conversations, and judge whether a generated reply reflects that user’s stylistic and pragmatic tendencies. With three tasks, ten domains, more than 111k conversation nodes, and thousands of personalized targets, this would be prohibitively resource-intensive. In addition, annotation quality would depend on annotators internalizing each user’s history, which is difficult to standardize. For these reasons, we prioritize a fully automated and reproducible evaluation framework for the first version of the benchmark.

---

### Official Review · Reviewer_Ux4W · 2025-11-01

**Soundness:** 3
**Presentation:** 2
**Contribution:** 2
**Rating:** 2
**Confidence:** 3

**Summary:**

This paper introduces PERSONACONVBENCH, a large-scale benchmark that evaluates how LLMs perform personalized reasoning and generation in multi-turn conversations. It integrates personalization and dialogue context across ten domains and three tasks—classification, regression, and text generation—showing that using user-specific conversation history improves model performance.

**Strengths:**

The paper constructs the first benchmark that jointly models personalization and multi-turn dialogue, enabling systematic evaluation of LLMs’ ability to adapt to user-specific styles and evolving conversational context.
By representing multi-user conversations as directed temporal graphs, the benchmark captures realistic branching, temporal ordering, and inter-user dependencies—allowing for fine-grained personalization and contextual reasoning that go beyond flat dialogue datasets.

**Weaknesses:**

Problem formulation lacks clarity:
The notation is underspecified — in particular, while Cu​ (the user trajectory set) is later defined, the meaning of f is not clearly introduced where it first appears. This makes it difficult to precisely understand what constitutes the model input.

Ambiguity in task setup and visibility scope:
It is unclear whether the model has access to all users’ conversational trajectories or only those of the participants in the current dialogue. In real conversations, a user A replying to B might also draw on prior interactions with other users (e.g., C). The paper does not explicitly explain whether such cross-thread context is included, how it is implemented. If implemented, whether temporal constraints are also enforced in such cross-thread context (i.e., that a reply at time t can only use information from ≤ t – 1). Without a clear temporal or visibility restriction, the need for a graph-based formulation is weakened.

Line 218: The phrase “conditioned on the conversational trajectory and the user’s trajectory set” is conceptually ambiguous. Are these two distinct inputs to the model, or do they represent different levels of abstraction of the same information? My understanding is that the conversational trajectory refers to an abstract notion, while the user’s trajectory set denotes the concrete collection of conversations associated with a specific user. If they are indeed separate inputs, please clarify their respective definitions, roles, and how they differ in practice.

Evaluation limitations:
For the dialogue generation task, it appears that each message has only one reference reply as ground truth. Metrics such as ROUGE are thus poorly suited to capture the diversity and open-endedness of conversational responses, limiting the reliability of quantitative evaluation.

**Questions:**

see above

---

> ### Author Response · Authors · 2025-11-26
> **Response (Part 1)**
>
> Thank you for the thoughtful comments.
>
> > **Formulation lacks clarity**
>
> Thank you for pointing out the ambiguity. We will revise the problem formulation to explicitly introduce all variables when they first appear. In particular, we now clarify that “f” refers to the structured input derived from the conversational trajectory “C” that contains the target message “m_tau”. This input corresponds to all earlier messages in that trajectory, written as: all messages m_t in C such that t_t < t_tau. The user trajectory set “C_u”, defined in Section 3.1, contains all trajectories that include messages authored by user “u”. This aligns with the problem formulation in Section 3, where the dataset is written as a collection of triples (f_i, y_i, C_{u_i}) for i = 1 to M, and “C_u” is explicitly described as the set of user-authored trajectories.
>
> This revision makes the model inputs fully explicit and resolves the notation ambiguity.
>
>
> > **Ambiguity in task setup and visibility scope**
>
> We agree that clarity on visibility constraints, temporal restrictions, and the scope of user-specific histories is essential for understanding why a graph-based formulation is required. We provide a detailed clarification below.
> ### **1. What conversational trajectories are visible to the model?**
> The model only has access to the trajectory set C_u belonging to the target user u.
> This is explicitly defined in Sec. 3.1 (“User Trajectory Set”) where C_u is described as:
> all trajectories that contain messages authored by user u,
> independent of the current thread,
> extracted from all posts initiated by u.
> Thus, **the model never sees the trajectories of other users**.
> ### **​​2. Are cross-thread interactions included?**
> **The benchmark only uses conversation trees in which the same user v is the root author**. Threads initiated by other users, even if user v participated in them, are not included in the trajectory set C_v.
>
> However, restricting the data to v-rooted threads does not simplify the problem or reduce the need for careful structural modeling.
> **Each thread rooted by user v naturally expands into large, multi-participant conversational trees:**
>
> (a) these trees include numerous branches,
>
> (b) each branch contains replies from many different users,
>
> (c) topic shifts and interactions unfold at different depths and timestamps,
>
> (d) reply dependencies form a non-linear structure instead of a linear dialog.
>
> Thus, even though all trajectories originate from the same user v, the conversation trees still represent rich multi-user interactions, often involving dozens of participants with different viewpoints and engagement patterns.
>
> This complexity is the central reason a graph-based formulation is required.
> The model must process:
>
> (a) branching conversational flows,
>
> (b) heterogeneous reply chains,
>
> (c) multi-level structures with large depth and width (Table 2),
>
> (d) temporal ordering across many interacting users,
>
> (e) the specific path that leads from the root post to the target reply.
>
> These structural properties remain present regardless of whether cross-thread participation is excluded.
>
> ### **3. Are temporal constraints enforced across these root-level threads?**
> Yes. The temporal rule applies strictly to all messages within all threads rooted by user v:
> Only messages with timestamps ≤ t_tau - 1 are included, regardless of which root-level thread they belong to.
> Thus, even within root-level threads:
>
> (a) No future message authored by v is visible.
>
> (b) No future message in any branch of the conversation tree is visible.
>
> The temporal ordering is globally respected across all data fed into the prompt.

---

> ### Author Response · Authors · 2025-11-26
> **Response (Part 2)**
>
> > **The phrase “conditioned on the conversational trajectory and the user’s trajectory set” is conceptually ambiguous.**
>
> ### **1. The two components are distinct inputs, not two abstractions of the same information**
> The conversational trajectory and the user’s trajectory set C_u are separate and complementary sources of information.
> They differ in:
>
> (a) their scope,
> (b) their construction rules,
> (c) their purpose in the benchmark,
> (d) and their visibility constraints.
>
> They are not two representations of the same data.
>
> ### **2. Conversational Trajectory (Local, One-Thread, Time-Ordered Path)**
> A conversational trajectory is:
>
> (a) a single time-ordered path within a tree-structured conversation,
> (b) extracted from the thread in which the target message m_tau appears,
> (c) containing only messages preceding m_tau in that same thread.
>
> It captures local interaction flow:
>
> (a) immediate preceding turns,
> (b) topic development,
> (c) reply dependencies,
> (d)the surrounding multi-user context relevant to that specific conversation.
>
> This is defined formally as a linear path
> C = (m1, ..., mT), where (mi, mi+1) ∈ E and ti < ti+1.
> (Ref. Sec. 3.1, “Conversational Trajectory”)
>
> ### **3. User’s Trajectory Set C_u (Global, Multi-Thread, All Root-Level Threads of the Same User)**
>
> The user’s trajectory set is:
>
> (a) the complete set of conversation trees rooted by the same user u,
> (b) collected across all posts initiated by u,
> (c) with strict temporal filtering (only content with ti mestamp < t_tau).
>
> This set does not include:
>
> participation in threads initiated by other users, and cross-thread replies where the user was not the root author.
> (Ref. Sec. 3.1, “User Trajectory Set”)
>
> ### **4. Why both components are necessary**
>
> The conversational trajectory provides the local flow of the discussion, allowing the model to follow the unfolding semantics, the structure of the reply chain, and the progression leading to the target message. In contrast, the user trajectory set C_u supplies long-term behavioral signals that shape how the user typically expresses sentiment, frames arguments, and writes replies. These long-range patterns are crucial for tasks such as sentiment classification, impact forecasting, and next-text generation. Using both sources of information jointly allows the benchmark to evaluate whether models can reason about the immediate conversational context while also adapting to stable user-specific tendencies: something that cannot be achieved with either component alone.

---

> ### Author Response · Authors · 2025-11-26
> **Response (Part 3)**
>
> > **Evaluation limitations**
>
> Thank you for the valuable feedback. We agree that conversational generation is inherently open-ended, and relying on a single ground-truth reply introduces challenges. Below we clarify why our current setup is methodologically sound.
>
> ### **1. The benchmark intentionally reflects the real structure of Reddit conversations**
>
> Each reply in our dataset corresponds to a specific human-written response in a multi-user conversation tree. This structure is intrinsic to the Reddit data distribution, where a user u produces exactly one reply to a given branch. Because our goal is to evaluate personalized next-text generation: predicting what this user would write in a concrete, observed context, having a single ground-truth reference is an accurate reflection of the task rather than a design choice that reduces diversity.
> The benchmark does not require predicting all possible plausible responses; it requires predicting the actual follow-up message produced by the same user, conditioned on earlier conversational dynamics and the user’s trajectory set.
>
> ### **2. Use of ROUGE and related lexical metrics follows established practice for single-reference evaluation**
>
> Although ROUGE and BLEU are limited for evaluating free-form generation with multiple valid answers, they remain standard metrics in existing work that evaluates models on grounded or personalized generation with a single reference (e.g., LaMP, LongLaMP, and prior conversation benchmarks).
>
> These metrics capture:
>
> (a) how closely the model aligns with the user’s actual writing pattern in that moment,
> (b) whether the generated text mirrors key phrases, structure, or intentions present in the true reply,
> (c) consistency in replicating the user’s style within a concrete conversational branch.
>
> For this benchmark, ROUGE does not attempt to score open-domain creativity; it approximates local alignment with the user’s actual response.
>
> ### **3. We explicitly address the lexical limitations by adding a semantic similarity metric**
> To mitigate the weaknesses of purely lexical metrics, the benchmark includes SBERT-based semantic similarity, which captures broader meaning-level alignment.
> This semantic metric evaluates: intent, thematic relatedness, and the overall semantic space of the reply.
>
> As shown in Table 3, SBERT provides a complementary perspective that is not sensitive to token-level overlap.
>
> ### **4. Why the evaluation remains meaningful even with a single ground truth**
>
> Although dialogue can support multiple plausible continuations, the task here is not unconstrained free-form generation. It is personalized, grounded in a specific conversational branch, and shaped by the user’s prior behavior within that thread. The observed reply represents how this user actually responded under that exact context, which makes it a valid supervision signal even if alternative continuations might also be reasonable. This supervised setup is consistent with established personalization benchmarks such as LaMP and LongLaMP, where each instance is paired with a single user-written output (e.g., subject line, long-form text) and models are evaluated against that reference using the same family of metrics. In our case, the aim is similarly to assess whether the model can approximate the user’s demonstrated behavior in a defined conversational situation, and measuring proximity to the observed reply remains meaningful for this purpose.

---

### Author Response · Authors · 2025-11-28
**Gentle Follow-up**

Dear Reviewers,

As we enter the final week of the discussion period, we would like to gently follow up. Our author response has been posted, and we remain available to address any further questions or clarifications. We appreciate your time and would welcome any additional feedback.

Best regards,

Authors

---

### Note · Authors · 2025-12-09

I have read and agree with the venue's withdrawal policy on behalf of myself and my co-authors.